# A detachable interface for stable low-voltage stretchable transistor arrays and high-resolution X-ray imaging

Yangshuang Bian[1,2], Mingliang Zhu[1,2], Chengyu Wang[1,2], Kai Liu[3], Wenkang Shi[1,2], Zhiheng Zhu[1,2], Mingcong Qin[1,2], Fan Zhang[1,2], Zhiyuan Zhao[1,2], Hanlin Wang [1,2], Yunqi Liu [1,2] & Yunlong Guo [1,2] ✉

Challenges associated with stretchable optoelectronic devices, such as pixel size, power consumption and stability, severely brock their realization in high-resolution digital imaging. Herein, we develop a universal detachable interface technique that allows uniform, damage-free and reproducible integration of micropatterned stretchable electrodes for pixel-dense intrinsically stretchable organic transistor arrays. Benefiting from the ideal heterocontact and short channel length (2 μm) in our transistors, switching current ratio exceeding $10^6$, device density of 41,000 transistors/$cm^2$, operational voltage down to 5 V and excellent stability are simultaneously achieved. The resultant stretchable transistor-based image sensors exhibit ultrasensitive X-ray detection and high-resolution imaging capability. A megapixel image is demonstrated, which is unprecedented for stretchable direct-conversion X-ray detectors. These results forge a bright future for the stretchable photonic integration toward next-generation visualization equipment.

Renovations in skin-like image sensors hold promise for intelligent visualization applications spanning from proactive healthcare and automatic identification systems, to augmented reality and soft robots[1–5]. Liberating radiation detectors from conventional flat-panel architectures is a crucial prerequisite to accurately sensing complex objects and ultimately capturing high-quality images in remote, continuous, and long-duration missions[6–8]. Serial studies on flexible X-ray detectors have been devoted to achieve the adjustable morphing that can effectively alleviate irradiation damage and image distortion[7–11]. Incorporation of the nanoscintillators[7], perovskites[8], or organic semiconductors[9] into soft materials has been proven an appealing mean for constructing flexible imagers, however, the further integration of such materials with intrinsically stretchable electrodes or even pixel arrays remains challenging given the lack of reliable production techniques[12–14]. Moreover, several essential specifications such as device miniaturization, mechanical deformation, and computational efficiency demand to be fulfilled toward fully extendable digital imaging[14–18].

Intrinsically stretchable organic transistors with excellent signal amplification and conformable integration capability have burgeoned from fundamental elastic logic components to wearable multifunctional appliances[13,19–23]. Sustained advancements, predominantly in material design[9,21,23–25], interface optimization[13,26,27], and manufacture technology[19,28,29], have made stretchable organic transistors readily available for elastic image sensors. For instance, skin-like multiplexed arrays have recently been exploited using stretchable organic transistors as the addressing unit[13] and pixel sensor[22], respectively. Despite prototypes for the curvature-adjustable imagers, intrinsically stretchable organic transistor-based image sensors are still in their infancy for obtaining high-resolution X-ray maps owing to the limited stability and integration. Indeed, direct microlithography techniques have currently enabled high levels of device density and uniformity for stretchable organic transistor arrays[19,29], but that are largely circumscribed by the specific molecular and complicated procedures. It should be noted that these transistors typically suffer from

[1]Beijing National Laboratory for Molecular Sciences, Key Laboratory of Organic Solids, Institute of Chemistry Chinese Academy of Sciences, Beijing 100190, China. [2]School of Chemical Sciences, University of Chinese Academy of Sciences, Beijing 100049, China. [3]Key Laboratory of Polymer Chemistry and Physics of Ministry of Education, School of Materials Science and Engineering, Peking University, Beijing 100871, China. ✉e-mail: guoyunlong@iccas.ac.cn

degradation of electrical performance and operational stability caused by the unclean or damaged interfaces in the fabrication process. Therefore, creating the desired heterointerface remains a long-standing issue for the photonic integration in skin-like electronics, which can guide high stability[30], ultrasensitive detection[21], and spatial resolution over the imaging[31,32].

Here we develop a universal stretchable photonic integration platform involving a detachable interface design to fabricate pixel-dense intrinsically stretchable organic transistors for high-resolution digital imaging. Our unique interface offers substantial advantages: damage-free micropattern, reproducible transfer, and ideal hetero-contact of intrinsically stretchable electrodes. Using this approach, we successfully integrated high-resolution stretchable electrodes for intrinsically stretchable organic transistor arrays with a high density up to 41,000 transistors/cm$^2$. Especially, the resultant devices simultaneously yielded a rather high switching current ratio ($I_{on}/I_{off}$) of ~10$^6$, the low-voltage operation of 5 V and excellent stability (including operational, long-term, and tensile stability), that could be attributed to the achieved short channel and ideal interface contact. Furthermore, we attained an outstanding photosensitivity and a high resolution from our stretchable transistor-based X-ray image sensors, which were comparable with the rigid direct X-ray imagers[33]. Notably, an integrated circuit element was imaged with a megapixel map by the stretchable organic X-ray image sensor, which was unprecedented for extendable direct-conversion digital imaging to our knowledge. As a result, our flexible photonic integration platform opens a facile and universal route toward densely integrated epidermal photonic devices and extendable digital imaging procedures.

## Results

### Design of the detachable interface for stretchable photonic integration

It is gradually recognized that flexible integration is a decisive step to stretchable organic transistors applicable for elastic pixel circuits and digital imaging technology[17,19,29]. Through suitable molecular and interface design, the developed reproducible transfer methodology has realized damage-free van der Waals integration of two-dimensional semiconductors on rigid substrates[34–38]. Nonetheless, reliable integration of intrinsically stretchable materials has yet to be proved, particularly the stretchable photonic integration of micropatterned stretchable electrodes, which is pivotal to scaling down the device size and simultaneously improving the operational stability and functionality. For this objective, an innovative detachable interface in the fabrication process of high-resolution stretchable electrodes is designed, allowing their damage-free micropattern and integration for stable intrinsically stretchable organic transistors (Fig. 1a and Supplementary Fig. 1). Figure 1a schematically depicts the production flow for densely integrated stretchable electrodes of carbon nanotubes (CNTs), in which a detachable interface is delicately constructed by depositing lithium fluoride (LiF) as the sacrificial layer (see Methods for details). Initially, inspired by the micropatterned metal electrodes through mature lift-off technology[39–41], we speculated that stretchable CNT electrodes could be directly deposited with high-resolution patterns defined by microlithography. Nevertheless, uncontrolled damage and unattainable transfer of the patterned CNT electrodes blocked their further accessibility to intrinsically stretchable optoelectronic devices (Supplementary Fig. 2a). To tackle this difficulty, the detachable interface was then designed to restrain the exfoliation-induced defects and assist subsequent transfer by introducing the LiF sacrifice layer between the micropatterned photoresist (PR) and CNT electrodes (Fig. 1a and Supplementary Fig. 2b).

This detachable interface strategy provides micropatterned stretchable electrode arrays with high uniformity, yield, and density, simultaneously enabling the dry transfer onto the elastic substrates. Figure 1b shows the high-resolution CNT patterns on both rigid and

stretchable substrates that can be utilized as source and drain electrodes with the channel length down to 2 μm. Highlighting the feasibility of flexible integration, we performed a micropatterned stretchable electrode array over an area of 2 × 2 cm$^2$, which could be further integrated with organic semiconductors toward the high-density intrinsically stretchable organic transistor matrix containing 164,000 devices (Supplementary Fig. 3). As showcased in Fig. 1c, the calculated density of stretchable organic transistor arrays reached as high as about 41,000/cm$^2$, comparable to the level realized by recently reported high-density elastic circuits[29]. Moreover, diverse micro-patterns of intrinsically stretchable electrode arrays were obtained that could be stretchable over 100% strain (Fig. 1d). Taken together, it suggests that the detachable interface approach can broaden the material choice and microfabrication technology, providing a brand-new pathway towards highly integrated stretchable functional electronics.

To elaborate on the detachable interface favorable for stretchable photonic integration, optical microscope and electrical characterizations of densely integrated intrinsically stretchable electrodes are first conducted. Relative to the conventional lift-off process with pristine interface, the assistance of the detachable interface offered high resolution and batch-to-batch repeatability for the micropatterned stretchable CNT electrodes (Supplementary Fig. 2). Furthermore, the intact transfer of these stretchable electrodes was achieved by introducing the detachable interface, which was well confirmed by the retained electrical conductivity and stretchability (Fig. 1e and Supplementary Fig. 4). Unfortunately, the LiF molecule had no obvious improvement on the stretchability and energy level of CNT electrodes due to the rather thin LiF layer after the transfer. Accordingly, we concluded that the insertion of LiF layer could enable pleiotropic effects: the coverage of the unclean substrate induced by photography and hence little damage in pre-fabrication process, a higher surface energy of the LiF/CNTs stack and ultra-high shear modulus[42] of LiF for the transfer, and the large membrane stress of the evaporated LiF layer for mechanical stripping (Supplementary Fig. 5 and Table 1). As for the possible mechanism between LiF and CNTs (–COOH groups), it might attribute to the Li–O dipole interaction that could reform quickly and thus enabling the reversible breakage and reformation for the deformation during the transfer process[43,44]. As a result, our detachable interface led to a low peeling force down to 0.3 N for readily integrating the high-resolution electrodes onto stretchable substrates (Fig. 1f), ensuring the ideal heterocontact without additional damage, impurities, and traps. Corroborated by the mechanical simulations of the pristine and LiF-assisted transfer process, the delamination of stretchable electrodes on the rigid substrate is achieved by the detachable interface with low Von Mises strain experienced between LiF and CNT layers, demonstrating the further transfer accessibility (Fig. 1g and Supplementary Movies 1 and 2).

### Electrical characterization of stable low-voltage stretchable organic transistor arrays

Advanced stretchable transistor technologies are generally driven by Moore's laws of shrinking the channel length, necessitating great improvements in four figures of merit: performance, power, uniformity, and stability[17]. Thus, a comprehensive assessment of as-fabricated intrinsically stretchable organic transistor arrays is of utmost significance for future high-end applications. The proposed intrinsically stretchable organic transistor employs a bottom-gate top-contact (BGTC) architecture with a channel width-to-length ratio ($W/L$) of 8 μm/2 μm, utilizing spatially nanoconfined hybrid polymer (poly[(2,5-bis(4-decyltetradecyl)-3,6-di(thiophen-2-yl)-2,5-dihy-dropyrrolo[3,4-c]pyrrole-1,4-dione)-co-(2,2′-dithiophen)] and styrene-ethylene-butylene-styrene elastomer blends, PDPP-BT/SEBS as the stretchable semiconductors, SEBS as the stretchable insulators and CNTs as the stretchable electrodes (Fig. 2a, b). The overall fabrication

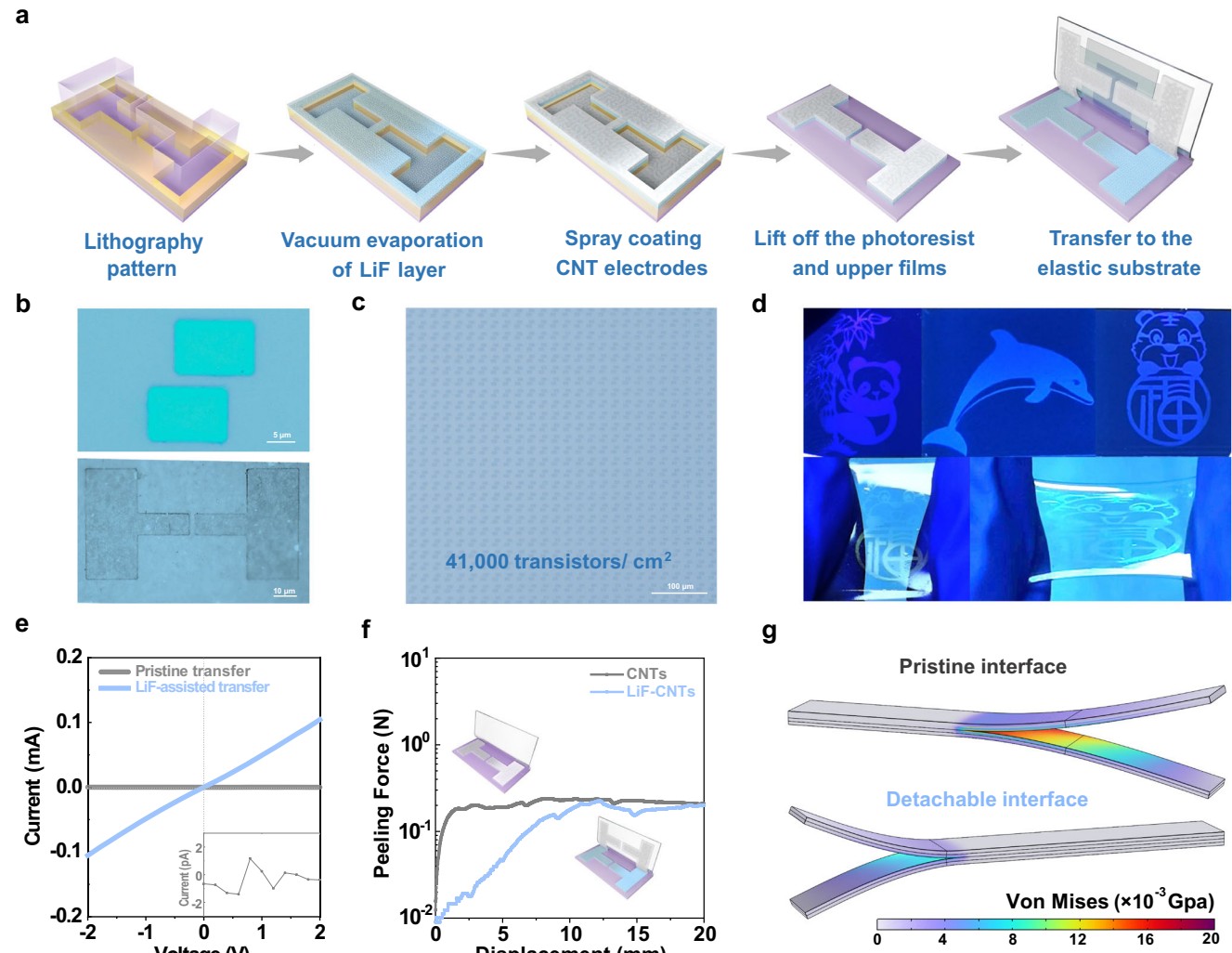

**Fig. 1 | Detachable interface design for the flexible integration of high-density stretchable organic transistor arrays. a** Schematic illustration of the production flow combining the detachable interface design and lift-off technology for densely integrated stretchable CNTs electrodes. **b** High-resolution optical microscope images of the micropatterned stretchable electrodes with feature size of 2 μm on Si-wafer (upper) and stretchable devices (bottom). **c** Optical microscope image showing the integrated intrinsically stretchable organic transistor matrix with a high density of 41,000/cm². **d** Photographic images of the fabricated stretchable CNT electrodes with different patterns. **e** Conductivity of the CNT electrodes transferred onto the stretchable substrate with and without the assistance of LiF layer constructed for the detachable interface. **f** The measured peeling force as a function of applied displacement in the transfer process of prefabricated CNT electrodes with pristine interface and detachable interface. **g** Mechanical simulation showing the Von Mises strain distribution during the peeling process of prefabricated CNT electrodes with pristine interface and detachable interface.

flow of highly integrated intrinsically stretchable transistor arrays is provided in the Methods and Supplementary Fig. 1. For the convenient measurement, we randomly extracted and measured the electrical performance of a 10 × 10 transistor array with a fixed channel length/width ratio of 2 μm/8 μm using large electrode pairs (Fig. 1b bottom). The resultant stretchable transistors showed an ideal switching behavior even at a low operating voltage of 5 V, including minimal current hysteresis, high on/off current ratio ($I_{on}/I_{off}$) exceeding $10^6$, low sub-threshold slope (*SS*) down to 317 mV per decade and high hole mobility ($μ_h$) of 0.580 cm² V$^{-1}$ s$^{-1}$ (Fig. 2c, d). In contrast to previously reported high-density stretchable organic transistors[29], the driving voltage and off-state current were reduced by ~5 fold and 3 orders of magnitude, respectively, that offered great potential toward multifarious on-skin photonic electronics. To verify the uniformity, we measured the transfer curves and extracted the charge-carrier mobilities of 100 transistors in a 10 × 10 array (Fig. 2e and Supplementary Fig. 6). An average $μ_h$ of 0.444 cm² V$^{-1}$ s$^{-1}$ with the corresponding standard deviation ($σ_μ$) of 0.0616 cm² V$^{-1}$ s$^{-1}$ was observed (Supplementary

Fig. 7). The narrow distribution of $μ_h$ indicates the reliable fabrication of high-density stretchable organic transistor arrays without significant performance difference. Furthermore, the overall device yield of 92.5% was achieved for the 100-device arrays in five different baches with few failed transistors owing to the leakage of the dielectric layer. Despite substantial efforts in stretchable semiconductors, high-permittivity materials, and fabrication technology, it remains challenging to realize overall electrical performance with high integration and low power consumption. Compared with previously reported stretchable organic transistors[9,19,23,25,28,29], our high-stability transistors simultaneously improved the *SS*, operating voltage, and integration scale (Fig. 2f). All these remarkable performances in our stretchable transistor array can be attributed to the short channel and high-quality hetercontact between electrodes and semiconductors achieved by the detachable interface method. It is noteworthy that the achieved low operating voltage and high operational stability can be attributed to the enhanced modulation of gate and reduced leakage current with the shorten channel length, respectively.

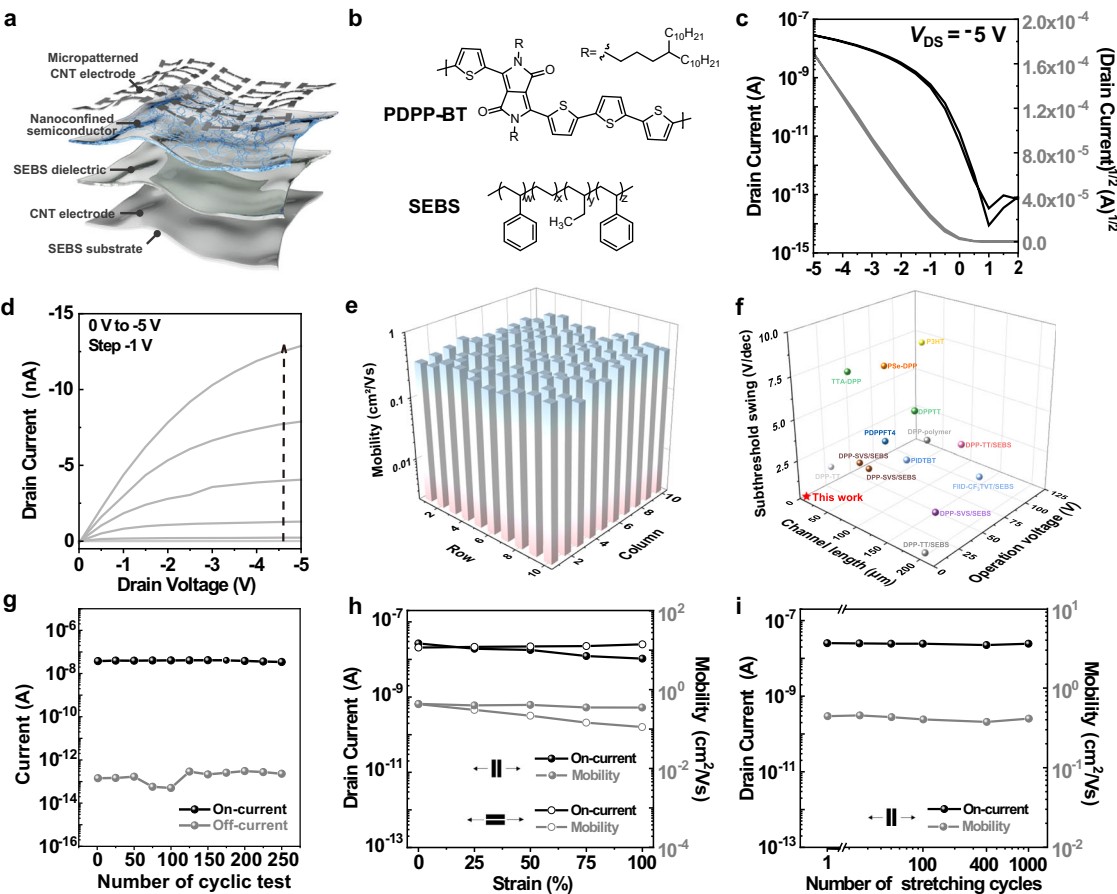

**Fig. 2 | Low-voltage-operated, high-stability intrinsically stretchable organic transistors. a** Schematic diagram illustrating the device configuration and constituent materials for high-density stretchable organic transistors. **b** Chemical structures of the PDPP-BT semiconducting polymer and SEBS elastomer. **c** Representative transfer curve from the short-channel stretchable organic transistor arrays ($W/L = 8\,\mu m/2\,\mu m$). The applied drain-to-source voltage denoted as $V_{DS}$, is $-5$ V. **d** Typical output characteristic showing the linear behavior at the small drain voltage region for the ohmic contact. **e** Histograms of mobility extracted from the $10 \times 10$ transistor array stochastically selected in the sheet. **f** Comparison of previously reported intrinsically stretchable organic transistors with our high-stability transistor in terms of the channel length, operation voltage, and subthreshold swing (see details in Supplementary Table 3). **g** Changes in the on- and off-current during 250 switching cycles. **h** On-current and mobility under different stretching levels parallel and perpendicular to the charge transport direction. **i** On-current and mobility obtained at 25% strain during 1000 stretching cycles parallel to the charge transport direction.

We further investigated the device stability including operational, long-term, tensile, and cyclic stability, all of which are essential parameters in remote and mobile missions. The operational stability was first validated by benchmarking the transfer characteristics of stretchable transistors subjected to 250 repeated switching cycles (Supplementary Fig. 8). The drain current values in both off and on states were well maintained during the cyclic tests, which was the first time to report high-stability intrinsically stretchable organic transistors to our knowledge (Fig. 2g). In terms of the long-term stability, the transistor properties of short-channel devices remained almost consistent after storage for 8 months (Supplementary Fig. 9). Moreover, the presented stretchable transistors exhibited high stretchability up to 100%, permitting excellent tensile stability beyond the static operation. Negligible variation in transfer characteristics at different tensile strains parallel or perpendicular to the channel direction could be observed (Supplementary Fig. 10). When applied tensile strain along the channel direction, the on-current value could occur a slight drop, while the estimated mobility mostly retained the initial value on account of the increased channel length during stretching (Fig. 2h and Supplementary Fig. 10a). As for the perpendicular direction, we found a slight reduction in the mobility, accompanied by the increased on-current (Fig. 2h and Supplementary Fig. 10b). Our stretchable transistor also showed highly stable electrical performance and excellent robustness in the 1000 cyclic stretching test at 25% strain along the

channel direction (Fig. 2i and Supplementary Fig. 11). Consequently, the low-voltage-operated, high-density intrinsically stretchable organic transistors with extraordinary stability are readily accessible to broadening applicability of wearable equipment.

## Ultrasensitive X-ray detection of densely integrated stretchable organic transistor arrays

As an initial proof-of-concept demonstration for fully extendable digital imaging procedures, the X-ray detection capability and irradiation stability of intrinsically stretchable organic transistor arrays were systemically investigated. The mass attenuation and attenuation efficiency of hybrid semiconductor materials were conducted to evaluate the intrinsic photon conversion efficiency to X-ray (Supplementary Fig. 12). In fact, unlike indirect detection, direct X-ray detectors based on transistor structures have a weak dependence of conversion efficiency because of the unique photo-modulation mechanism involving the charge accumulation by deep trap states, thereby resulting in highly sensitive response of such low-Z organic materials under ionizing radiation[9,45,46]. Therefore, our stretchable transistor-based organic X-ray image sensors exhibited ultrasensitive X-ray response owing to the achieved short channel length, high carrier mobility, and low dark current, even with a low absorption of hybrid semiconductor films (Fig. 3a and Supplementary Fig. 12). The observed photoswitching behaviors, including the ascending current

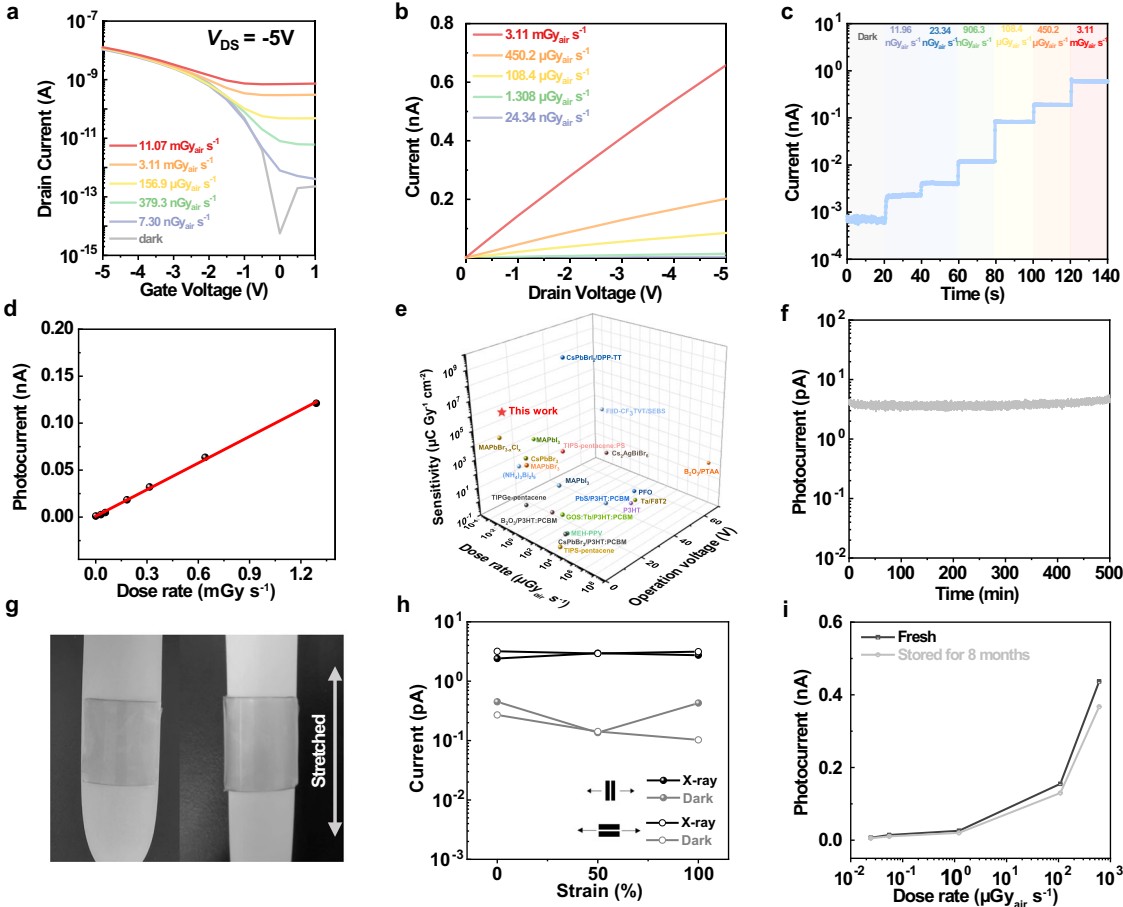

**Fig. 3 | Ultrasensitive detection performance of intrinsically stretchable transistor-based organic X-ray image sensors. a** Transfer characteristics of the device measured in the dark and irradiation for the intrinsically stretchable organic transistor ($W/L = 8\,\mu m/2\,\mu m$) when biased at $V_{DS} = -5\,V$. **b** Drain current versus voltage curves measured under X-ray irradiation at different dose rates. **c** Dynamic response of the stretchable transistor-based X-ray detector with −5 V bias voltage ($V_{DS}$) at the off-state ($V_G = 0.5\,V$). **d** X-ray induced photocurrent versus dose rate in a vacuum (<100 Pa). **e** Performance comparison including the detection limit, operation voltage, and sensitivity of current reported X-ray detectors with our prepared stretchable X-ray sensors in this work (see details in Supplementary

Table 4). **f** X-ray induced photocurrent drift of the intrinsically stretchable X-ray detector for 500 min under 24.34 nGy$_{air}$ s$^{-1}$ dose rate, biased at $V_{DS} = -5\,V$ and $V_G = 0.5\,V$. **g** Visualized conformability and ultra-flexibility of the stretchable X-ray image sensors wrapped around a water balloon under stretching. **h** X-ray photocurrent and dark current versus strain parallel and perpendicular to the charge transport direction under 24.34 nGy$_{air}$ s$^{-1}$ dose rate, biased at $V_{DS} = -5\,V$ and $V_G = 0.5\,V$. **i** Long-term detection stability of the short-channel intrinsically stretchable organic transistor stored for 8 months in a glove box under different dose rates at $V_{DS} = -5\,V$ ($V_G = 0.5\,V$).

values of source-to-drain current ($I_{DS}$) and positive shifts of threshold voltage ($V_{th}$), demonstrated the photogating effect on tuning the photocarrier trapping in the semiconducting channel (Fig. 3a). However, the weak photogating effect was observed at the high gate voltage ($V_G$) region that could be ascribed to the near-saturated current at a high intrinsic gate field and the fast recombination of photo-generated carriers in our short-channel phototransistors, which was common in the previously reported phototransistors. Figure 3b shows $I_{DS}$ as a function of the source-to-drain voltage ($V_{DS}$) under different dose rates of the X-ray irradiation. Obviously, $I_{DS}$ increased monotonically with $V_{DS}$, that was because the high electric field between drain and source electrodes could promote the extraction of photo-generated charges in the channel. To further demonstrate the dynamic X-ray response, the device was serially exposed with different irradiation intensity stepped up from 11.96 nGy$_{air}$ s$^{-1}$ to 3.11 mGy$_{air}$ s$^{-1}$ (Fig. 3c). It was clearly to see that our stretchable X-ray image sensors enabled a wide detection range, stable photocurrent, and rapid response characteristics (Fig. 3c and Supplementary Fig. 13).

To exclude the air-ionization contribution to the devices, we also conducted the control experiments under the vacuum and air conditions to confirm the actual performances of the prepared transistor-

based detectors (Supplementary Fig. 14). Indeed, the background current could influence the photocurrent at the low dose rate region, so we further measured the photoresponse in a vacuum (<100 Pa) to obtain the accurate sensitivity of our stretchable short-channel transistors (Supplementary Fig. 15). However, the similar photocurrent at high dose rates in both vacuum and air conditions might be attributed to the sufficient free charges from the semiconductor, source/drain electrodes and the presence of air. As shown in Fig. 3d, the well linear scaling of photocurrent was observed with the increase of irradiation intensity in a vacuum. Next, we calculated the sensitivity of the device that reached a high value up to $5.74 \times 10^6\,\mu C\,Gy^{-1}\,cm^{-2}$ (Fig. 3d, e). Noted that the degraded performance at dark was caused by the associated changes in series resistance, leakage current, and ambient noises from the connected wires and vacuum equipment (Supplementary Fig. 16).

Considering the scalable pixel-dense integration, the performance uniformity and stability of stretchable organic X-ray detector arrays deserve further attention. The map and statistical histograms of photocurrent showed a minimal fluctuation among the 100 counted devices, proving the reliable detection capability of our stretchable transistor-based X-ray detector arrays (Supplementary Fig. 17). In the

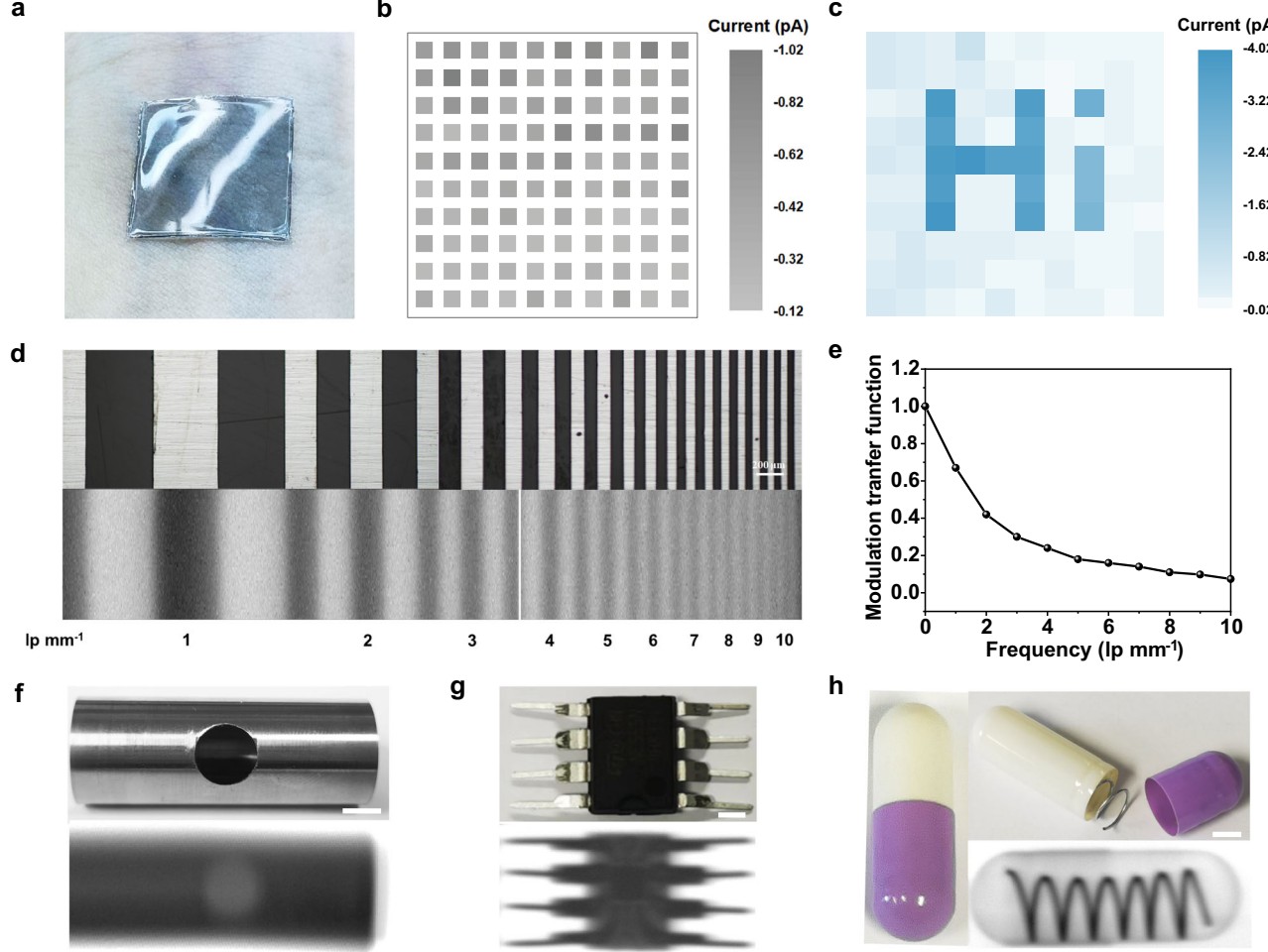

**Fig. 4 | High-resolution X-ray imaging based on intrinsically stretchable organic image sensors. a** Images showing the skin-like characteristics of the intrinsically stretchable organic transistor-based image sensor arrays. **b** Dark current of stretchable organic X-ray image sensors in a $10 \times 10$-pixel array. The applied $V_{DS}$ and $V_G$ are $-5$ V and 0.5 V, respectively. **c** Map of the X-ray response for the word "Hi" at a low dose rate down to 24.34 $nGy_{air} s^{-1}$. **d** Optical microscope (upper) and X-ray images (bottom) of a molybdenum plate patterned with alternating fringes of different line pairs (lp) (The scanning spacing is 5 μm). **e** MTF of the intrinsically stretchable organic X-ray image sensor. **f** Photograph (upper) and corresponding X-ray image (bottom) of the steel sheet with a round hole using the short-channel stretchable organic transistor (2 μm) (The scanning spacing is 200 μm). **g** Photograph (upper) and corresponding X-ray image (bottom) of an integrated circuit device (The scanning spacing is 15 μm). **h** Photograph, and corresponding X-ray image of an encapsulated metallic spring (The scanning spacing is 20 μm). The scale bar is 2 mm.

air measurement, the lowest detection limit of 24.34 $nGy_{air} s^{-1}$ was the actual detectable dose rate with a signal-to-noise ratio (SNR) value of 4.85 for X-ray irradiation (Supplementary Fig. 18). Additionally, a fast response speed was also observed in the short-channel device including the rise time of ~0.2 s and fall time of ~1.7 s at the dose rate of 24.34 $nGy_{air} s^{-1}$ (Supplementary Fig. 19). We reasoned that these excellent results could be ascribed to the low dark current of ~$10^{-13}$ A and high electric field of $2.5 \times 10^4$ V $cm^{-1}$ between the source and drain electrodes enabled by the high-quality interfacial contact and short channel. Additionally, the photocurrent of the device showed nearly no degradation even under irradiation for 500 min (Fig. 3f) that indicated intriguing potential for long-life usage at a low dose rate. Toward future on-skin photonic electronics, we probed the stretchability and long-term stability of the stretchable X-ray image sensors as well. The tissue-level softness of our stretchable image sensors make it possible to conformally adhere to irregular objects that largely improve the resolution and accuracy of real-time signals (Fig. 3g). When subjected to 100% strain, our device exhibited a relatively stable photoresponse that could represent the high detection stability during stretching (Fig. 3h and Supplementary Fig. 20). Notably, the X-ray response of the stored image sensor remained constant over eight months (Fig. 3i),

suggesting the outstanding long-term stability of the detector that was integral to high-resolution X-ray imaging procedures.

## High-resolution imaging of intrinsically stretchable transistor-based image sensors

With the reduction of device size and operational voltage, the ultra-sensitive intrinsically stretchable transistor-based X-ray detectors seem to well gratify the desire for extendable high-resolution X-ray imaging. The imaging experiments of objects were mainly performed by our single-pixel imaging system through recording a series of signals from individual exposed short-channel stretchable device that was related to coordinates of the moved objects (Supplementary Fig. 21). We also measured the device one by one to demonstrate the excellent imaging capability of the proof-of-concept stretchable image matrix. As illustrated in Fig. 4a, our intrinsically stretchable detector matrix could be conformally attached onto the human wrist, minimizing the discomfort reactions and detection bias for bioelectronics. Achieving intrinsically stretchable image sensor arrays with stable and low dark current is a primary requisite for high-quality X-ray imaging. Figure 4b shows the mapping of dark current values extracted from 100 devices at $V_{DS} = -5$ V and $V_G = 0.5$ V. The mean values for $I_{dark}$ and $J_{dark}$ were

$2.9 \times 10^{-13}$ A and $1.8 \times 10^{-6}$ A cm$^{-2}$, respectively, which typically were on par with the state-of-the-art stretchable X-ray detectors[9]. Taking advantage of the ultralow dark current, we produced a simple image of "Hi" at an ultralow dose rate of 24.34 nGy$_{air}$ s$^{-1}$ that was suitable for further application in wearable scenario (Fig. 4c). To clarify the resolution of our transistor-based detectors for extendable X-ray imaging, a molybdenum plate patterned with alternating fringes of different line pairs was performed (Fig. 4d). The spatial frequency, that is, the pairs of lines per millimeter (lp mm$^{-1}$) reached the highest value of 10 lp mm$^{-1}$ for our short-channel device, which was the first stretchable organic image sensor reported for high-resolution direct-conversion X-ray imaging. The modulation transfer function (MTF) is determined by the maximum ($I_{max}$) and minimum brightness ($I_{min}$) in the relevant line pair (see Methods for the detail calculation). From the MTF curve, the spatial frequency of 5 lp mm$^{-1}$ was obtained when the MTF declines to 0.2 (Fig. 4e). Noted that the MTF values are generally not relevant to the dose rate of X-ray irradiation[12].

To illustrate the generality of stretchable organic image sensors in the industrial, electronic, and medical scenario, we verified their imaging capability via visualizing the structure and content of the examined objects. For comparison, we detected a round hole in the stainless-steel pipe using the stretchable transistor-based X-ray image sensor with different channel lengths of 2 μm and 200 μm, respectively. Relative to the large-size transistor prepared by conventional methods[9,47], our short-channel device offered more pixel numbers, enabling a more detailed view of the pipe (Fig. 4f and Supplementary Fig. 22). To confirm that stretchable image sensors pervaded well in nondestructive inspection, an electronic component of the integrated circuit was experimented. The resulting image was palpably clear to resolve the inner connects of base pins with about $1.1 \times 10^{6}$ pixels (Fig. 4g), which was unprecedented for extendable direct-conversion digital imaging. As a showcase for high-contrast X-ray imaging, we further imaged a closed capsule with a metallic spring concealed inside (Fig. 4h). The hidden spring was clearly discerned owing to the large difference in X-ray attenuation of plastic and metal materials. Also, benefiting from the ultrasensitive X-ray detection and enough small pixel size of our stretchable organic image sensors, the overlapped wall of white and purple parts in the capsule could be clearly visualized. To further illustrate the flexible superiority, we conducted the imaging comparison of a pipe with a square hole on the top using the rigid and stretchable short-channel transistor array, respectively (Supplementary Fig. 23). Unlike the rigid detector, our stretchable transistor array could be inserted and attached into the pipe, thus achieving a high-contrast image. However, the rigid flat-panel detectors might need even higher irradiation to penetrate the pipe to obtain similar quality of images. The results of single-pixel imaging and array imaging depend on the operational stability of the individual device and device-to-device uniformity, respectively. All imaging results demonstrate the great potential of our short-channel stretchable transistor array for practical applications. The high resolution and accuracy of our image sensors make them available to deeply process and analyze the information, facilitating the development of stretchable sensor technology in intelligent medical imaging.

## Discussion

We introduce an innovative design of the detachable interface in the fabrication process for stable and high-density intrinsically stretchable organic transistor-based image sensors. Our detachable interface enables high-resolution micropattern and reproducible integration of intrinsically stretchable electrodes that are readily applicable to stretchable photonic integration. Utilizing this technique, the fabricated transistors achieve significant improvements in device density (41,000 transistors/cm$^2$), operational stability (>250 consecutive cycles), and electrical performance (switching current ratio exceeding $10^6$ at a low operating voltage of 5 V). Moreover, we validate their

ultrasensitive X-ray detection capability with a high photosensitivity and a lowest detection limit down to 24.34 nGy$_{air}$ s$^{-1}$. Furthermore, the intrinsically stretchable organic image sensors achieve high-resolution X-ray imaging and a megapixel imaging, which are unprecedented for stretchable direct-conversion image sensors. Therefore, our approach provides a valuable platform to densely integrated skin-like photonic systems for future artificial centralizing applications.

## Methods

### Materials

The styrene ethylene butylene styrene elastomers (SEBS-H1221 and SEBS-H1052) were purchased from Asahi Kasei (Japan). SEBS-H1221 with volume fractions of 88% poly(ethylene-co-butylene) was used as the stretchable substrates and secondary elastic component in the stretchable hybrid semiconductors. SEBS-H1052 with volume fractions of 80% poly(ethylene-co-butylene) was employed to fabricate the stretchable dielectric layer. Single-walled carbon nanotube (P3-SWNT) was supplied by Carbon Solutions (America) for the fabrication of gate, drain, and source electrodes. P-type conjugated polymer (PDPP-BT) with the number-averaged molecular weight of 68.4 kDa and polydispersity of 2.4 Đ was synthesized from our laboratory (Supplementary Fig. 24). PR (AZ5214) was purchased from Merck (Germany). LiF was purchased from LUMTECH Taiwan Corp. All solvents, including toluene, chlorobenzene, acetone, and cyclohexane were supplied by J&K Seal.

### Film and device fabrication

Each stretchable functional layer except for the source and drain electrodes were prefabricated on the OTS-modified substrates. The polymer semiconductor films were prepared by depositing the hybrid semiconductor solution (PDPP-BT: SEBS-H1221 = 3:7, 10 mg ml$^{-1}$ in chlorobenzene) on the OTS-modified Si/SiO$_2$ wafers with a spinning speed of 2000 rpm and an annealing temperature of 150 °C for 10 min. The gate electrodes were prepared by spray-coating the P3-SWNTs solution (0.25 mg mL$^{-1}$ in isopropanol) on the OTS-modified Si/SiO$_2$ wafers at 120 °C with a distance of 8–12 cm between the spray gun and the wafer. The dielectric layers were coated on the OTS-modified Si/SiO$_2$ wafers with a spinning speed of 3500 rpm, and followed by annealing at 80 °C for 1 h. The SEBS-1052 solution was 65 mg mL$^{-1}$ in cyclohexane, and the resultant thickness of dielectric layer was about 1200 nm that was similar with the literature[47] (Supplementary Table 2). The substrate layers were prepared on the OTS-modified glass sheet by dropping the SEBS-1052 solution (200 mg mL$^{-1}$ in toluene), followed by annealing at 40 °C for 5 h. The thickness of substrate was approximately 800 μm.

As shown in Fig. 1a, the source and drain electrodes were fabricated using lift-off technology with a detachable interface design. First, the PR was spin-cast and photolithographically patterned on the cleaned Si/SiO$_2$ wafer to form the PR template. Then, the small molecule of LiF was evaporated on the top of wafer with a thickness of 100 nm to construct the detachable interface. Next, CNT electrodes were spray-coated on the stack. After the deposition of electrodes, the PR layer with upper CNTs was removed by dissolving in the acetone with a ultrasound power of 50 W for 1 min, while high-resolution stretchable electrode array was obtained. Finally, the electrodes can be easily transferred onto stretchable substrates by mechanical stripping.

The intrinsically stretchable organic transistor arrays with BGTC configuration were fabricated by sequentially transferring the gate electrode, SEBS dielectric, polymer semiconductor and micro-patterned source/drain electrodes (channel width/length = 8/2 μm) by the SEBS substrate from their Si/SiO$_2$ wafers (Supplementary Fig. 1). The gate, dielectric, and semiconductor layers could be easily transferred by thermal lamination method in a vacuum oven at 50 °C for 30 min. The micropatterned source/drain electrodes can be directly

transferred onto the as-fabricated device stack by mechanical stripping (Fig. 1).

## Characterization

Optical microscope images were measured by a BX53 cross-polarized optical microscope. The contact angle was obtained from Drop Shape Analyzer (DSA100, KRÜSS) by calculating the average value of the left and right angles of a sessile drop in static mode based on the tangential method. The 180° peel-off measurement was carried out by universal testing machine (KJ-10653B, China). The characterization of all the films took place at room temperature in ambient atmosphere.

## Optoelectrical measurements

The electrical performance was performed by the Keithley 4200 in vacuum conditions. The X-ray detection characteristics were measured by the PDA FS380 in ambient atmosphere using Moxtek MAD-PRO (60 kV 12 W) as the X-ray source. The height and width of focal spot generated from the X-ray beam were 1.26 mm and 0.30 mm, respectively. The distance between the detector and X-ray source was 20 cm, and a 0.25 mm thickness of beryllium attenuator was inserted in the interspace. The attenuated dose rates were calibrated by the X-ray dosimeter (Model: MagicMax, IBA, German) under different current and voltage of X-Ray tube.

## Imaging measurements

The imaging capability characterized by a single-pixel system that yielded images through interrogating an object with a series of position-related patterns while recording the corresponding response value of the detector. The single-pixel imaging experiments were carried out on the scanning imaging platform (CAV/SPRATDS-2, Shanghai Yinxin Technology Corp.) connecting with a stepping-motor controlled translation (X–Y) stage and a signal sampling system in air. The imaged objects were fixed on the stepping-motor stage with a distance of ~0.5 mm between the object and the device. The system recorded the corresponding response value of the individual device according to the coordinates that were determined by moving the object along the x/y axis, respectively. Limited by the object size and scanning numbers of our single-pixel imaging system, the scanning distances of the line pair plate, steel sheet, integrated circuit, and encapsulated metallic spring were designed by 5 μm, 200 μm, 15 μm and 20 μm, respectively. The final images were recorded and processed by the imaging verification software (CAV1.1.5). As for the imaging experiments of stretchable and rigid stretchable transistor array, we measured the exposed devices in the transistor array one by one (Supplementary Fig. 22).

## Data availability

The authors declare that all data supporting the findings of this study are available within the paper and its Supplementary Information. Source data are provided with this paper.

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

## Acknowledgements

This work was financially supported by the National Key R&D Program of China (2018YFA0703200 (Y.G)), the National Natural Science Foundation of China (61890940 and U22A6002 (Y.G)), the Strategic Priority Research Program of CAS (XDB30000000 (Y.L.)), the CAS Project for Young Scientists in Basic Research (YSBR-053 (Y.G)), the CAS-Croucher Scheme for Joint Laboratories, Lu Jiaxi international team (GJTD-2020-02 (Y.G)), the CAS Cooperation Project (121111KYSB20200036 (Y.G)), and the Beijing Nova Program (20220484173 (Y.G)).

## Author contributions

Y.B. and M.Z. contributed equally to this work. Y.G. and Y.L. proposed and supervised the project. Y.G. and Y.B. conceived the idea. Y.B. designed experiments. M.Z. synthesized the semiconductor. C.W., W.S., Z. Zhao, and H.W. conducted the lithography experiments. K.L. and M.Q. conducted the fabrication of stretchable devices. Z. Zhu and F.Z. were involved in the mechanical characterizations. Y.G. and Y.B. wrote the manuscript and all authors reviewed it.

## Competing interests

The authors declare no competing interests.
