## [Peer Review File · Nature Communications]

Reviewers' comments:

Reviewer #1 (Remarks to the Author):

In this work, the authors have introduced a method for consistently integrating stretchable electrodes, enhancing the capabilities of inherently stretchable organic transistor arrays. With their optimal transistor designs, they achieved a high switching current ratio, high device density, low operational voltage, and reasonable stability. Their sensors demonstrate sensitive X-ray detection and high-resolution imaging capabilities. While the study is interesting and worthy of consideration for publication in Nature Communications, the following concerns must be addressed.

1. Authors state device density of 41,000/cm². However, the statistics are only shown for 100 devices. I would revise the claim and state the number of devices fabricated and measured instead of stating the device density. It can be misleading for the readers.
2. The yield for these devices is not mentioned.
3. Also I do not understand how the imaging experiments were performed. Were individual devices exposed one by one or all at a time?
4. Authors should also report the responsivity, detectivity, and response time for these phototransistors. What is the mechanism of phototransduction? How does the device-to-device variation influence the imaging outcome?

Reviewer #2 (Remarks to the Author):

This paper introduces an interesting technique for transferring high-density electrode arrays onto a target layer, thereby manufacturing an organic transistor array with shared gate electrodes. This development might contribute to the advancement of flexible devices and organic integrated circuits. However, the mechanism of the X-ray detector manufactured using this technique in this study remains much unclear. The design is not optimized according to the technical specifications of the X-ray detector. Essentially, it may be a micro-ionization chamber array that detects X-ray using air rather than the organic semiconductor. The sensitivity calculated based on the erroneous assumptions and methods leads to unreasonably high value. Compared to X-ray detection, using this device as a substitute for Si-based TFT arrays or for imaging under visible light may have more potential. But the author did not show how the elasticity can contribute to a better imaging application scenario.

In summary, I hold a very reserved opinion regarding the publication of this article. My main concerns lie in the research motivation and the reliability of the data. I will list my concerns as followings:

1. In this article, the X-ray absorption of the photosensitive layer is less than 0.01% (see Supplementary Figure 12), but an unreasonably high sensitivity of up to $10^9 \mu\text{C Gy}^{-1} \text{cm}^{-2}$ was achieved. This sensitivity surpasses that of state-of-the-art direct X-ray detectors by over three orders of magnitude. The simple attribution of this phenomenon to photoconductive

gain appears to be unreasonable. The author did not calculate the gain based on first principles. Instead, it was derived by comparing the measured sensitivity with the theoretical limit of sensitivity, which constitutes a form of circular reasoning.

2. Furthermore, I am skeptical about the presence of a significant gain effect in the sensor. As depicted in Figure 3a, with an increase of the gate voltage, the dark current undergoes a change of several orders of magnitude, while the corresponding X-ray response current remains nearly unchanged. If there were indeed gain, we would expect the X-ray response current at the same gate voltage to increase by a similar order. However, Figure 3a suggests that the applied gate voltage merely introduces a fixed amount of dark current, exerting minimal positive influence on the X-ray response. Moreover, it is noteworthy that the maximum signal-to-noise ratio for X-ray response is observed at a gate voltage of 0, which contradicts the photogating effects.

3. X-ray currents were tested with a gate voltage of 0.5V and VDS of -5V. Given the channel length of only $2\mu\text{m}$, the electric field between the source and drain terminals reaches up to 2500 V/mm. Even under such a high electric field, at an X-ray dose rate of 3 mGy/s, the absolute value of the response current is merely 600 pA (refer to Figure 3b). This stands in contrast to the claimed ultrahigh sensitivity. It is highly likely that the response current of this X-ray detector is primarily produced by air rather than the organic photosensitive layer. Surprisingly, this aspect is not addressed in the text. Notably, conventional X-ray detector ionization chambers utilize air as the detection material. Previous studies have shown similar instances of mistakenly attributing the response signal of air to the material's response, particularly in detectors with extremely small electrode areas, close channels, such as the ones showed in this manuscript. Hence, in this study, whether calculating the current density using the photosensitive area of the organic layer or the electrode area of the source and drain terminals, it consistently underestimates the current cross-sectional area, consequently inflating the calculated sensitivity. Assuming an average ionized air cross-sectional area of 1 mm^2 around your actual device, when you calculated with a device area in the range of square micrometers, the sensitivity obtained may be artificially inflated by nearly six orders of magnitude.

4. The sensitivity was measured at high electric field and low X-ray dose rate. This may not accurately reflect the true X-ray detection capability of the detector. When making comparisons with other detectors, conditions such as electric field strength should be standardized. The article mentions that the short channel leads to advantages in terms of higher electric field strength, thereby enhancing sensitivity and photocurrent. However, this also comes with negative effects such as increased dark current. At the end of the day, it is the SNR that matters rather than the sensitivity.

5. The minimum detection limit is claimed to be $24.34\text{ nGyair s}^{-1}$, but it does not explain how this measurement was obtained. More comprehensive data should be provided.

6. The dose rate for X-ray stability testing is only $24.34\text{ nGyair s}^{-1}$, which is too low for typical X-ray imaging scenarios. Additional stability data at higher dose rates should be included. And it is the total received (absorbed) dose that determines the material degradation under X-ray, given that the organic layer can only less 0.01%, the stability is significantly overestimated.

7. In the X-ray imaging demonstration, the data read from the line pair plate is 10 lp/mm, while the MTF is calculated at 5 lp/mm. This raises questions about the consistency of the claimed

resolution. Furthermore, based on the actual imaging demonstration of objects, it appears that this value has not been achieved.

8. In my understanding, the highlight of the article lies in the stretchable detector array. However, there is a lack of demonstration in imaging applications (all demonstrations are with rigid detectors). There is great anticipation to witness experimental outcomes for flexible or elastic X-ray imaging, and to explore how these imaging concepts may surpass conventional approaches. Moreover, there appears to be a logical gap: while the introduction suggests potential benefits of flexible or even elastic detector arrays, the ultimate imaging is still reliant on planar detector arrays.

Reviewer #3 (Remarks to the Author):

This manuscript presents a universal detachable interface design for fabricating high-density organic transistor arrays. The incorporation of LiF layer enables the precise patterning of CNT stretchable electrodes, seamlessly integrating them as source and drain electrodes within the organic transistor array. The authors have demonstrated a significantly higher density of stretchable organic transistor arrays with shorter channel lengths, lower operating voltages, and superior on/off ratios compared to the current state-of-the-art intrinsically stretchable organic transistor arrays.

Overall, this work represents a substantial advancement in the realm of stretchable electronic devices. I recommend its publication in Nature Communications, pending certain necessary revisions.

Here are some specific comments

(1) It remains unclear why the authors chose to use a LiF layer in their approach. It would greatly enhance the manuscript if the authors could provide experimental data elucidating the underlying mechanisms of this choice.

(2) With the exception of the LiF interface technique, the materials used in this study appear to be almost same with those reported in a previous reference (<https://www.science.org/doi/10.1126/science.aah4496>). Given the similarity in the fabrication procedures, it is essential for the authors to present additional characterization data of the semiconducting and dielectric layers. Supplementary Table 2 in this manuscript is almost identical to Table S3 in above reference. This is a little bit weird.

(3) To gain a deeper understanding of the low operating voltage and high operational stability observed in the study, it is imperative that the authors provide insights into the underlying mechanisms. Furthermore, it would be valuable to explore whether the use of LiF contributes to improved stretching behaviors of CNT electrodes and, if so, the rationale behind this improvement should be addressed.

(4) The manuscript showed the demonstration of a stretchable active matrix for X-ray imaging. However, it lacks a detailed description of the actual device structure and the fabrication process.

Responses (R) to the Comments (C)

Responses to the reviewer 1

In this work, the authors have introduced a method for consistently integrating stretchable electrodes, enhancing the capabilities of inherently stretchable organic transistor arrays. With their optimal transistor designs, they achieved a high switching current ratio, high device density, low operational voltage, and reasonable stability. Their sensors demonstrate sensitive X-ray detection and high-resolution imaging capabilities. While the study is interesting and worthy of consideration for publication in Nature Communications, the following concerns must be addressed.

R: We sincerely appreciate the reviewer's positive and insightful comments, and giving us the opportunity to further improve our manuscript. We carefully revised the manuscript and addressed all the concerns of the reviewer in the revised manuscript. The point-to-point answers for reviewer's comments are included in the following parts.

C1: *Authors state device density of 41,000/cm². However, the statistics are only shown for 100 devices. I would revise the claim and state the number of devices fabricated and measured instead of stating the device density. It can be misleading for the readers.*

R1: We appreciate the reviewer's valuable comments and we apologize for the confusion. In fact, we successfully fabricated an intrinsically stretchable short-channel transistor array containing 164,000 transistors within a 4-cm² substrate (Fig. 1c and fig. S3), enabling a high device density of ~41,000 transistors/cm². However, such high-density array with very small electrodes and a large number of devices made it difficult to connect with the probes to measure all the transistor performance. Therefore, we randomly extracted and measured the photoelectrical performance of a 10×10 transistor array with a fixed channel length/width ratio of 2 μm/8 μm using large electrode pairs (Fig.1b bottom) for convenient testing. As suggested, we have added the corresponding explanation in the revised manuscript to clarify this point to avoid any misleading interpretation to the reader (Page 5, line 5 and 6, Page 7, line 14 and 16).

C2: *The yield for these devices is not mentioned.*

R2: We appreciate the reviewer's helpful comments and we apologize for the omission. In fact, we presented the best performance of a 10×10 transistor array in the manuscript. As suggested, we supplemented the overall device yield of 92.5% that was achieved for the 100-device arrays in five different batches with few failed transistors owing to the leakage of the dielectric layer. In the revised manuscript, we have added the corresponding sentences to explain the device yield (Page 8, line 1 to 3).

C3: *Also I do not understand how the imaging experiments were performed. Were individual devices exposed one by one or all at a time?*

R1: We thank the reviewer for pointing out this issue and we apologize for the misleading. Indeed, the main imaging experiments were performed by our single-pixel imaging system through recording a series of signals from individual exposed short-channel stretchable device that was related to coordinates of the moved objects

(Methods, Page 20, line 6 to 13). It is similar to expose patterns of the object one by one to the individual device. As for the supplementary imaging experiments, we measured the exposed devices in the stretchable transistor array one by one. In this revised manuscript, we added the relevant explanations (Page 14, line 2 to 5) and the figure of single-pixel imaging system in the Supplementary Information (Fig. S21).

Figure R1. Schematic illustration of the single-pixel imaging system and the practical mounting boards connected with our stretchable transistors.

C4: Authors should also report the responsivity, detectivity, and response time for these phototransistors. What is the mechanism of phototransduction? How does the device-to-device variation influence the imaging outcome?

R4: We thank the reviewer for arising these issues, and giving us the opportunity to further improve our manuscript. In fact, we failed to obtain the responsivity and detectivity due to the unavailable intensity of incident light for the X-ray irradiation that can only be calibrated by dose rate. According to the previously reported references, we calculated the sensitivity that was generally used to evaluate the photo-response characteristics of transistor-based X-ray detectors under different dose rates (*Nat. Commun.* 2020, 11, 2136.; *Nat Commun.* 2016, 7, 13063; *Adv. Mater.* 2019, 31, 1901644). As the reviewer's suggested, we measured the response time including the rise time and fall time that were calculated about 0.2 s and 1.7 s, respectively. We have added the relevant explanation (Page 13, line 9 to 11) in the revised manuscript and the figure in the Supplementary Information (Fig. S19).

Figure R2. Temporal response (a) and high-resolution temporal response (b, c) of the intrinsically stretchable organic transistor under $24.34 \text{ nGy}_{\text{air}} \text{ s}^{-1}$ dose rate at $V_{\text{DS}} = -5\text{V}$ ($V_{\text{G}} = 0.5 \text{V}$).

As for the mechanism of phototransduction, it is still unclear and unattainable about the photoconductive gain mechanism and characterization of organic semiconductors for the X-ray detection to date. However, to address the reviewer's request, we further investigated the previous literatures and found that the mainstream perception of X-ray

detection was similar to the traditional photoelectric conversion process that involves the photoconductive gain effect. When the high-energy radiation exposures onto the organic semiconductors, the energy is absorbed by electrons through the photoelectric effect or Compton scattering, forming high-energy electrons, and thus leading to electron–hole pairs separation. High-efficiency carrier transport of active materials and appropriate electric field can directly convert the incident photons into a photocurrent and facilitate the electron–hole pairs migrate to their respective electrodes (*Sci. Adv.* 2021, 7, eabf4462.; *Nat. Commun.* 2020, 11, 2136.; *Nat Commun.* 2016, 7, 13063). The photogenerated electrons are easily captured by the deep traps at the low dose rate, thereby resulting in the increased carrier lifetime and higher photosensitivity. As the irradiation intensity increased, the deep traps are filled and then carriers gradually fill into shallower traps with smaller gain, which was recognized as the dynamic-range enhancing gain compression (*Nat. Commun.* 2023, 14, 6865; *Nat. Photon.* 2017, 11, 726–732.; *ACS Photon.* 2014, 1, 936–943). In this revised manuscript, we have added the relevant explanation (Page 10, line 18 to 20 and Page 13, line 2 to 6).

Figure R3. Schematic of the process of modulation of the conductivity induced by X-rays exposure of the semiconductors.

As for the influence of device-to-device variation on imaging outcome, the contrast and noise of images were usually affected by the device uniformity, sensitivity and stability. Indeed, the device-to-device dark- and photo- current of our stretchable transistor array had a small fluctuation that could induce subtle noise, but was almost invisible or negligible because of the relatively stable and sensitive response (Fig. 3f, 4b and Fig. S15). In particular, the single-pixel imaging demonstration mainly depend on the stability of the individual X-ray detector, enabling the high quality of imaging results (Fig. 4f-h). We have added the relevant explanation in this revised manuscript (Page 16, line 12 to 14).

Responses to the reviewer 2

This paper introduces an interesting technique for transferring high-density electrode arrays onto a target layer, thereby manufacturing an organic transistor array with shared gate electrodes. This development might contribute to the advancement of flexible devices and organic integrated circuits. However, the mechanism of the X-ray detector manufactured using this technique in this study remains much unclear. The design is not optimized according to the technical specifications of the X-ray detector. Essentially, it may be a micro-ionization chamber array that detects X-ray using air rather than the organic semiconductor. The sensitivity calculated based on the erroneous assumptions and methods leads to unreasonably high value. Compared to X-ray detection, using this device as a substitute for Si-based TFT arrays or for imaging under visible light may have more potential. But the author did not show how the elasticity can contribute to a better imaging application scenario.

In summary, I hold a very reserved opinion regarding the publication of this article. My main concerns lie in the research motivation and the reliability of the data. I will list my concerns as followings:

R: We sincerely appreciate the reviewer for the helpful and insightful comments, which are essentially important to improve our manuscript. We understand the reviewer's concern about the air-ionization and the obtained high sensitivity from the thin organic semiconductors. However, the mainstream X-ray detector with CZT materials mainly employed the heterojunction diode and Schottky-type detector, in which the thickness determined the dark current and photoelectronic conversion efficiency (*J. Mater. Sci.: Mater. Electron.* 2016, 27, 645–650.). However, the thickness or absorption may be not a major role in the phototransistors that are completely different from the vertical stacked detectors (*Sci. Adv.* 2021, 7, eabf4462.; *Nat. Commun.* 2020, 11, 2136.; *ACS Nano* 2019, 13, 6973–6981). Additionally, the previous literatures attributed to the excellent photo-modulation to the charge accumulation by deep trap level during X-ray exposure, thereby resulting in the inner amplification of signal current provided by the photoconductive gain mechanism (*Sci. Adv.* 2021, 7, eabf4462.; *Nat. Commun.* 2020, 11, 2136.; *Nat Commun.* 2016, 7, 13063). Specifically, in the OFET-based X-ray detectors the charge transport occurs at only a few nanometers (1–3 monolayers) adjacent to the dielectric interface rather than the bulk of the organic layer in the vertical devices (*Nat. Commun.* 2020, 11, 2136.; *Chem. Rev.* 2020, 120, 5, 2879–2949). Therefore, the air-induced photocurrent on the top could not directly affected the threshold voltage and transfer curves that depend on the inner carriers for the transistor with bottom-gate-top-contact structure.

Furthermore, the imaging demonstrations obtained from our short-channel intrinsically stretchable organic transistor have obvious improvements than the previously reported organic X-ray detectors (*Nat. Commun.* 2022, 13, 7163.; *Nat Commun.* 2016, 7, 13063.). Different from the rigid detector, stable and high-contrast imaging could be achieved by our short-channel stretchable transistors even under deformation. For the reviewer's convenience, the details are included in the point-to-point answers to the comments below.

C1: In this article, the X-ray absorption of the photosensitive layer is less than 0.01% (see Supplementary Figure 12), but an unreasonably high sensitivity of up to $10^9 \mu\text{C Gy}_{\text{air}}^{-1} \text{cm}^{-2}$ was achieved. This sensitivity surpasses that of state-of-the-art direct X-ray detectors by over three orders of magnitude. The simple attribution of this phenomenon to photoconductive gain appears to be unreasonable. The author did not calculate the gain based on first principles. Instead, it was derived by comparing the measured sensitivity with the theoretical limit of sensitivity, which constitutes a form of circular reasoning.

R1: We thank the reviewer for pointing out this issue, which is of crucial significance to improve our manuscript. As mentioned above, in the OFET-based X-ray detectors, the charge transport occurs at only a few nanometers adjacent to the dielectric interface rather than the bulk of the organic layer in the vertical devices. The absorption of photosensitive layers plays a minor role in the phototransistors that are completely different from the vertical stacked detectors (*Nat. Commun.* 2020, 11, 2136.; *Adv. Mater.* 2021, 33, 2101717.; *ACS Nano* 2019, 13, 6973–6981.). For clarity, a simple schematic diagram is provided below.

Figure R4. Schematic illustration of the detection process in the OFET-based X-ray detectors.

For the transistor-based X-ray detectors, the efficiency of carrier transport largely affected the separation and extraction of the photogenerated hole-electron pairs (*Nat. Commun.* 2020, 11, 2136.). Furthermore, we carefully consulted the relevant literatures and found that the sensitivity is related to the dose rate of irradiation and electric field intensity (*Nat. Commun.* 2023, 14, 6865; *Nat. Photonics* 2019, 13, 602–608.; *Adv. Mater.* 2021, 33, 2101717). Therefore, benefiting from the achieved short channel (largely reduced about five orders of magnitude), low noise and high mobility, our devices exhibited a high transport efficiency, and thus enabling a reasonable high sensitivity at such low dose rate of 24.34 nGy s^{-1} and high electric field intensity of $2.5 \text{ V}/\mu\text{m}$. In particular, the similar values or exponential growth with the decrease of dose rate have also been observed in some literatures (*Adv. Mater.* 2021, 33, 2101717; *Nat. Commun.* 2020, 11, 2136.; *Adv. Mater.* 2019, 31, 1901644.).

We also apologized for misleading the reviewer about the photoconductive gain. In fact, previous literatures attributed to the photo-modulation of the semiconductor conductivity to the charge accumulation during X-ray exposure, thereby resulting in the inner amplification of signal current provided by the photoconductive gain mechanism (*Sci. Adv.* 2021, 7, eabf4462.; *Nat. Commun.* 2020, 11, 2136.; *Nat Commun.* 2016, 7, 13063). A simple schematic was provided to describe the process of photoconductive gain effect as follows. Generally, holes drift along the electric field while electron was

trapped. When the holes reached the collecting electrode, the mobile hole continuously re-injected from the injecting electrode to guarantee charge neutrality. Consequently, when exposed to X-rays, each photogenerated electron-hole pair could induce more than one hole to contribute the photocurrent, ultimately leading to a photoconductive gain effect.

Figure R5. Schematic of the process of modulation of the conductivity induced by X-rays exposure of the semiconductors.

Here, we calculated the sensitivity and gain according to the reference (*Nat. Commun.* 2023, 14, 6865.; *Nat. Photon.* 2017, 11, 726–732; *Nat Commun.* 2016, 7, 13063; *Adv. Mater.* 2019, 31, 1901644.). The calculated gain includes the photoconductive gain and impact ionization gain, it was calculated by the ratio of the response current and theoretical current, so the gain is relatively independent on the sensitivity that derived by the slope of current-dose rate. However, the two parameters were both related to the photocurrent and dose rate and used to evaluate the photoresponse characteristics of transistors. Here, we ascribed the outstanding values of gain and sensitivity to the low dark current of $\sim 10^{-13}$ A, high mobility of $0.444 \text{ cm}^2 \text{ V}^{-1}$ and high electric field of $2.5 \times 10^4 \text{ V cm}^{-1}$ between the source and drain electrodes enabled by the achieved high-quality interfacial contact and short channel (Page 13, Line 11 to 14). To avoid any misunderstanding, we have revised the relevant explanation in this manuscript (Page 10, Line 18 to 20).

As mentioned about the calculation of gain based on first principles, we fully agree the reviewer that it's an effective approach to avoid any form of circular reasoning and provide more valid results. Unfortunately, it is difficult to realize owing to the unclear mechanism in the photoconductive process. Additionally, the previously developed kinetic model (*Nat. Commun.* 2016, 7, 13063.) was not available for our device. The suggestions proposed by this reviewer are of crucial significance, we will deeply investigate the reliable calculation of gain in future.

C2: Furthermore, I am skeptical about the presence of a significant gain effect in the sensor. As depicted in Figure 3a, with an increase of the gate voltage, the dark current undergoes a change of several orders of magnitude, while the corresponding X-ray response current remains nearly unchanged. If there were indeed gain, we would expect the X-ray response current at the same gate voltage to increase by a similar order. However, Figure 3a suggests that the applied gate voltage merely introduces a fixed amount of dark current, exerting minimal positive influence on the X-ray response. Moreover, it is noteworthy that the maximum signal-to-noise ratio for X-ray response is observed at a gate voltage of 0, which contradicts the photogating effects.

R2: We thank the reviewer for arising this issue, and giving us the opportunity to better elucidate the correlation between the gate voltage, photocurrent and gain. As above

mentioned, the photoconductive gain effect was common in both resistor- and transistor-based photodetectors and X-ray radiation detectors (*Nat. Commun.* 2023, 14, 6865.; *Sci. Adv.* 2021, 7, eabf4462.; *Nat. Photon.* 2017, 11, 726–732; *Nat. Commun.* 2020, 11, 2136.). Crucial for the amplification in this mechanism is the slow recombination of X-ray generated carriers by trapping the free electron carriers from the recombination process. In brief, the photogating effect is caused by trapped photo-induced charges that modulate the potential energy of the semiconductor/dielectric interface, where these trapped charges contribute an additional electrical gating-field, resulting in a shift in the threshold voltage (*Nanomaterials* 2023, 13, 882.; *Sci. Adv.* 2021, 7, eabf4462.; *Adv. Sci.* 2017, 4, 1700323.). Thus, the trap state, carrier mobility and source-drain voltage play the major roles on X-ray response rather than the bulk gate voltage. To clarify this issue, we provided a simple schematic of the equivalent circuit in the process of modulation as follows.

Figure R6. Schematic of the detector under irradiation. The trapped electrons at the semiconductor/dielectric interface induce an increase of holes, thereby improving the semiconductor conductivity that is similar to applied an additional gate voltage (V_{photo}).

At the V_{G} of 0.5~1V (cut-off state), the photogating field caused by the photoconductive gain mechanism has a strong influence on the current of our transistor. However, at a high gate voltage (on-state mode), the weak photogating effect was reasonably observed due to the fast recombination of photo-induced carriers in our short-channel phototransistors and near-saturated current at a high intrinsic gate field (*Adv. Sci.* 2017, 4, 1700323.). In particular, the additional photogate field was proportional to the irradiating carriers and the induced current (from pA to nA) was reasonably negligible compared with the near-saturated current (~10 nA) at high intrinsic gate field, thereby showing minimal positive influence on the photocurrent. The similar phenomenon was also observed in the reported phototransistors based on various semiconductor (*Adv. Mater.* 2021, 33, 2101717.; *Nat. Commun.* 2020, 11, 101.; *Nat. Commun.* 2018, 9, 4546.). Therefore, the strong photogating effect was typically observed at the cut-off state, for our transistor, that was at V_{G} of 0.5 ~ 1V. Indeed, the obvious shift of the transfer curves (related to the threshold voltage) could directly confirm the photogating effect (“using photo as gate electrode to switch the transistor”) (Fig 3a). However, to avoid any possible misunderstanding to the reader, we have added the relevant explanation in this manuscript (Page 11, Line 1 to 4).

C3: X-ray currents were tested with a gate voltage of 0.5V and V_{DS} of -5V. Given the channel length of only $2\mu\text{m}$, the electric field between the source and drain terminals reaches up to 2500 V/mm . Even under such a high electric field, at an X-ray dose rate of 3 mGy/s , the absolute value of the response current is merely 600 pA (refer to Figure 3b). This stands in contrast to the claimed ultrahigh sensitivity. It is highly likely that the response current of this X-ray detector is primarily produced by air rather than the organic photosensitive layer. Surprisingly, this aspect is not addressed in the text. Notably, conventional X-ray detector ionization chambers utilize air as the detection material. Previous studies have shown similar instances of mistakenly attributing the response signal of air to the material's response, particularly in detectors with extremely small electrode areas, close channels, such as the ones showed in this manuscript. Hence, in this study, whether calculating the current density using the photosensitive area of the organic layer or the electrode area of the source and drain terminals, it consistently underestimates the current cross-sectional area, consequently inflating the calculated sensitivity. Assuming an average ionized air cross-sectional area of 1 mm^2 around your actual device, when you calculated with a device area in the range of square micrometers, the sensitivity obtained may be artificially inflated by nearly six orders of magnitude.

R3: Thank you for your comments. Actually, we have considered the effects of the air ionization to the X-ray photocurrent. As mentioned above, in the OFET-based X-ray detectors the charge transport occurs at only a few nanometers adjacent to the dielectric interface rather than the bulk of the organic layer in the vertical devices. Our devices have a bottom-gate-top-contact configuration, thus the air-induced photocurrent on the top is unlikely to affect the threshold voltage and transfer curves that depend on the inner carrier transport adjacent to the dielectric interface. However, to address the reviewer's concern, we provided the results of the control experiments as follows.

Figure R7. a. Schematic illustration of the detection process in the OFET-based X-ray detectors. b. I-V curves of the air and our transistor device at $11.07\text{ mGy}_{\text{air}}\text{ s}^{-1}$ dose rate ($V_{DS} = -5\text{ V}$). c. The imaging demonstration of the as-proposed micro-ionization chamber array with air as photoactive layers.

We confirmed that the probes connected our short-channel intrinsically stretchable transistors and measured the semiconducting characteristics (Fig. 3a and R8b). Although we recorded a non-zero signal from the “dummy” devices probably due to the air ionization effect induced by the secondary electrons emitted from the metal probes. However, the higher noise of dark current and lower photoresponse were observed in air compared with our transistor device. Therefore, we assume that the contribution of the air ionization is far smaller than the intrinsic devices. Furthermore,

it is also impossible to achieve high-resolution imaging by the air owing to the whole ionization of space rather than a small pixel provided by our organic transistors. We reckon that this is the most direct evidence to assess the contribution to the X-ray photocurrent from the air ionization. To elucidate it clearly, we have added a figure in supplementary information (Fig. S13), that shows the photocurrent signal and image result of a dummy device upon exposure to X-ray radiation. The data clearly shows that the photocurrent density of the air ionization is about five order of magnitude lower than that of the hybrid polymer semiconductors.

Referring the relevant literatures, the sensitivity is directly related to the current density and detectable dose rate (*Nat. Commun.* 2014, 5, 4174; *Nat. Photonics*, 2017, 11, 726–732; *Adv. Mater.* 2019, 31, 1901644). In our device, the achieved short channel not only induced the high electric field but also largely improved the high current density and detectable limit. In particular, we reduced the whole active area (channel size) by about nearly five orders of magnitude (from 200 $\mu\text{m}/4000 \mu\text{m}$ to 2 $\mu\text{m}/8 \mu\text{m}$), while largely maintaining the photoresponse. Therefore, the resultant high current density, high carrier mobility and low detectable limit of our stretchable transistor together led to the reasonable high sensitivity at the electric field of 2.5 V/mm.

We also fully understand the reviewer’s concern that the definition of current density can significantly affect the sensitivity of the X-ray detectors. Indeed, we measure the typical transistor characteristics inside the device rather than the external response of air ionization (Figure R7a and b). The current density is defined as the current which is flowing through one unit value of a cross-sectional area. Referring to the reported X-ray detectors based on a TFT geometry, the pixel area used for the calculation of the sensitivity is the total area of charge channel (*Adv. Funct. Mater.* 2019, 29, 1806119; *Adv. Mater.* 2021, 33, 2101717.; *Nat. Commun.* 2020, 11, 2136.). The expression is depicted as follows:

$$J = \frac{I}{WL}$$

where J is the current density, and I is the current. W and L represent the width and length of the channel, respectively. To avoid any misunderstanding, a schematic diagram of the channels area (blue color) is provided below.

Figure R8. Schematic illustration of the channel area of the STOFET-based X-ray detectors.

Furthermore, assuming an average ionized air cross-sectional area of 1 mm^2 , the contribution of air-induced photocurrent density ($\sim 0.1 \text{ nA}/\text{mm}^2$) to the channel region (2 $\mu\text{m}/8 \mu\text{m}$) was far lower than the inner photocurrent density of our transistor ($\sim 6.25 \times 10^4 \text{ nA}/\text{mm}^2$). To explain it intuitively, we have added the relevant explanation in this manuscript (Page 11, line 4 to 8).

C4: *The sensitivity was measured at high electric field and low X-ray dose rate. This*

may not accurately reflect the true X-ray detection capability of the detector. When making comparisons with other detectors, conditions such as electric field strength should be standardized. The article mentions that the short channel leads to advantages in terms of higher electric field strength, thereby enhancing sensitivity and photocurrent. However, this also comes with negative effects such as increased dark current. At the end of the day, it is the SNR that matters rather than the sensitivity.

R4: We are grateful for the reviewer's helpful comments and constructive suggestions. The electric field, dark current density and SNR of X-ray detectors are really critical parameters that need to be considered for X-ray detectors. According to reviewer's comments, we added the calculation of dark current density and the figure of SNR value in the revised manuscript (Page 13, line 9 and 12), and revised the table (Supplementary Table 3) to present a systematic comparison with the reported X-ray photodetectors.

Table R1. Comparison in the performance of the prepared short-channel intrinsically stretchable organic transistor with the previously reported X-ray detectors.

The type of devices	Photoactive materials	Structure	Operate Voltage (V)	Electric Field (V/ μm)	X-ray source (anode voltage/ X-ray energy)	Minimum dose rate (mGy _{air} s ⁻¹)	S _A ($\mu\text{C Gy}_{\text{air}}^{-1} \text{cm}^{-2}$)	Ref.
Perovskite detectors	MAPbBr _{3-x} Cl _x	Diode	-5	2.5×10 ^{-3*}	--; 8keV	7.60×10 ⁻⁶	8.4×10 ⁴	24
	CsPbBr ₃	Diode	1.2	5×10 ^{-3*}	W; 50keV	3 ×10 ⁻²	5.57×10 ⁴	25
	MAPbBr ₃	Diode	-1	5×10 ^{-4*}	Cu; --	3.6×10 ⁻²	2.1×10 ⁴	26
	MAPbI ₃	Diode	30	0.03	W; 50kV	2.2×10 ⁻⁷	1060	27
	MAPbI ₃	Resistance	10	0.333*	Cu; 8keV	0.928	650*	28
	(NH ₄) ₃ Bi ₂ I ₉	Resistance	10	6.5×10 ^{-3*}	Ag;50keV	5.50×10 ⁻⁵	803	29
	Cs ₂ AgBiBr ₆	Resistance	50	0.025*	W; 50kV	5.2×10 ⁻³	105	22
Direct organic inorganic hybrid X-ray detectors	B ₂ O ₃ /P3HT:PCBM	Diode	-10	0.5*	W; 50kV	0.13	4.79*	30
	Ta/F8T2	Diode	-50	2.5*	Mo;17.5keV	5	0.217*	31
	B ₂ O ₃ /PTAA	Diode	-100	20*	Mo;17.5keV	13	0.15*	
	PbS/P3HT:PCBM	Diode	-30	0.067*	W; 40kV	10	3.66*	32
	CsPbBr ₃ /DPP-TT	Transistor	-60	1.5*	--; 20 kV	1.0×10 ⁻³	3×10 ⁹	20
Indirect organic inorganic hybrid X-ray detectors	GOS:Tb/P3HT:PCBM	Diode	-10	10	W; 70kV	1.5	7.35*	33
	CsPbBr ₃ /P3HT:PCBM	Diode	-3	0.2	Cu; 40–80 kV	58.18	3.67	34
Direct organic X-ray detectors	PFO	Diode	-50	2.5*	Mo; 50keV	4	0.96*	35
	MEH-PPV	Diode	-10	0.5*	Mo; 50keV	4	0.48*	
	P3HT	Diode	44	1.517*	--	16.6	0.47*	36
	TIPS-pentacene	Resistance	0.2	0.057*	Mo;17 keV	55	0.77*	37
	TIPGe-pentacene	Transistor	-3	0.067*	Mo; 35 kV	6.4×10 ⁻³	18*	19
	TIPS-pentacene:PS	Transistor	-20	0.8*	Mo; 35 kV	3.5×10 ⁻²	1.3×10 ⁴	21
	FIID-CF ₃ TVT:SEBS	Transistor	60	0.3*	Ag; 20 keV	3.77×10 ⁻⁵	1.52×10 ⁴	15
	DPP-BT/SEBS	Transistor	-5	2.5	Ag; 7 kV	2.4×10 ⁻⁵	3.78×10 ⁹	This work

*Values extracted by the device information reported in the referenced papers.

The expression of signal-noise ratio (SNR) is depicted as follows:

$$SNR = \frac{\bar{X}_i}{\sigma_{X_i}}$$

where \bar{X}_i and σ_{X_i} are the signal current and noise current, respectively. The signal current is defined as the average value of photocurrent (X_i)²⁰:

$$\bar{X}_i = \langle X_i \rangle = \frac{1}{n} \sum_{i=1}^n X_i = \frac{1}{n} \sum_{i=1}^n (x_n^{i'} - \overline{x_{dark}})$$

where $\overline{x_{dark}}$ and x_n^i are the average value of dark current and data point of signal current, respectively.

As for the noise current is determined by the standard deviation of signal current.

$$\sigma_{\bar{X}_i}^2 = \langle (X_i - \bar{X}_i) \rangle = \frac{1}{n} \sum_{i=1}^n (X_i - \bar{X}_i)^2$$

Figure R9. Signal-to-noise ratio of the short-channel stretchable transistor-based detector derived by calculating the standard deviation of the X-ray signal current under different dose rates at $V_{DS} = -5V$ ($V_G = 0.5 V$). The red dashed line represents a SNR of 3.

In accordance to the reviewer's request, we have added the equation and the figure of SNR values (Fig. S18) in the Supporting information (Section 3, Page 14).

C5: The minimum detection limit is claimed to be $24.34 \text{ nGy}_{air} \text{ s}^{-1}$, but it does not explain how this measurement was obtained. More comprehensive data should be provided.

R5: We thank the reviewer for pointing out this issue and we apologize for the confusion. Indeed, the minimum detection limit of $24.34 \text{ nGy}_{air} \text{ s}^{-1}$ was the actual detectable dose rate. The dose rate was calibrated by the X-ray dosimeter (Model: MagicMax, IBA, German) under different current and voltage of X-Ray tube. And the attenuated dose rates were further calculated through the absorption of 0.25 mm thickness of beryllium attenuator. Additionally, the lowest detection limit of $24.34 \text{ nGy}_{air} \text{ s}^{-1}$ was also at the edge of the actual measurement with a SNR value of 4.85 (Supplementary Fig. 18). In this revised manuscript, we added the related explanation (Page 13, line 8 and 9).

C6: The dose rate for X-ray stability testing is only $24.34 \text{ nGy}_{air} \text{ s}^{-1}$, which is too low

for typical X-ray imaging scenarios. Additional stability data at higher dose rates should be included. And it is the total received (absorbed) dose that determines the material degradation under X-ray, given that the organic layer can only less 0.01%, the stability is significantly overestimated.

R6: We sincerely appreciate the reviewer for the helpful and insightful comments. Indeed, organic detectors with low noise and high sensitivity present a great potential in the X-ray imaging at a low dose rate. Although the obtained stability is not suitable for applications in high energy-discriminated X-ray spectral imaging, it is feasible for the digital mammography at a low dose rate with non-damage imaging (*Nat Commun.* 2016, 7, 13063; *Adv. Mater.* 2019, 31, 1901644.; *Sci. Adv.* 2021, 7, eabf4462.). As mentioned, the stability under high-energy X-ray irradiation, it deserves to be investigated in future owing to the possible material damage and degradation of organic polymer semiconductors. In this manuscript, we mainly emphasize the high stability of our device including the air, operational, stretching and cyclic stability (Fig. 2g-i). To address the reviewer's concern, we revised the excessive explanation to state the irradiation stability at a low dose rate in the revised manuscript (Page 13, line 16).

C7: *In the X-ray imaging demonstration, the data read from the line pair plate is 10 lp/mm, while the MTF is calculated at 5 lp/mm. This raises questions about the consistency of the claimed resolution. Furthermore, based on the actual imaging demonstration of objects, it appears that this value has not been achieved.*

R7: We sincerely appreciate this reviewer for pointing out the data from the line pair plate, and apologize for the misleading interpretation. In fact, we demonstrated the resolution of our stretchable devices using the single-pixel imaging system. Although the image of 10-line pairs was achieved, it was not clear. MTF is a common parameter to evaluate the imaging capability of the image sensor at different spatial frequencies. It is generally considered that the imaging quality is good when MTF is higher than 0.2. From the MTF curve, the spatial frequency of 5 lp mm⁻¹ was obtained when the MTF declines to 0.2, which was comparable to the reported high-resolution X-ray detectors (*Nat. Electron.* 2021, 4, 681–688.). However, to address the reviewer's concern, we have deleted the inappropriate statements in this revised manuscript to avoid any misunderstanding in the revised manuscript (Page 1, 3 and 17).

As for the actual imaging demonstration of objects, the different test conditions resulted in the resolution deviation of the imaged objects. Indeed, to obtain the best resolution of our short-channel stretchable devices, the line pair plate with a relatively small size was imaged with a minimum scanning distance of 5 μm. However, the actual objects with large size are difficult to image at the same condition due the limitation of the scanning numbers of our single-pixel imaging system. Therefore, the scanning distances of the steel sheet, integrated circuit and encapsulated metallic spring were 200μm, 15μm and 20μm according to their object sizes, respectively. To clarify this issue, we provided the detailed measurement conditions of various imaging demonstration in the revised manuscript (Methods, Page 20, line 7 to 12).

C8: *In my understanding, the highlight of the article lies in the stretchable detector array. However, there is a lack of demonstration in imaging applications (all*

demonstrations are with rigid detectors). There is great anticipation to witness experimental outcomes for flexible or elastic X-ray imaging, and to explore how these imaging concepts may surpass conventional approaches. Moreover, there appears to be a logical gap: while the introduction suggests potential benefits of flexible or even elastic detector arrays, the ultimate imaging is still reliant on planar detector arrays.

R8: We sincerely appreciate this reviewer for pointing out the witness experimental outcomes for flexible or elastic X-ray imaging. In fact, all the images were obtained based on our short-channel intrinsically stretchable organic transistors using our single-pixel imaging system that working through moving objects with scanning the single stretchable devices. To explain it clearly, we provided the schematic of the single-pixel imaging system and the practical imaging connection of our stretchable transistors. We also added the figure (Fig. S21) in the Supplementary Information.

Figure R10. Schematic illustration of the single-pixel imaging system and the practical mounting boards connected with our stretchable transistors.

Different from the rigid detector, stable image was also can be achieved under deformation. However, concerning reviewer's comments, we supplemented the simple comparison of imaging demonstrations based on flexible and rigid transistor arrays as follows. As the figure shown below, the hole can be clearly imaged when the detector array was bent and inserted into the pipe. However, the poor resolution or contrast was observed when placing the detector array outside the pipe, resulted from the larger attenuation as the X-ray irradiation need penetrate two layers to the detectors. To obtain the similar quality of image, rigid flat-panel detectors need even higher irradiation. In this revised manuscript, we supplemented the figure (Fig. S23) and relevant explanation to demonstrate the superiority of our X-ray detectors (Page 10, line 7 to 12 and Page 20, line 13 to 14).

Figure R11. Schematic (left) and experimental results (right) of the hole of the pipe in the wall imaging by the stretchable detectors from inside (a) and the rigid detectors from outside (b) (All images were taken without noise subtraction).

Responses to the reviewer 3

In This manuscript presents a universal detachable interface design for fabricating high-density organic transistor arrays. The incorporation of LiF layer enables the precise patterning of CNT stretchable electrodes, seamlessly integrating them as source and drain electrodes within the organic transistor array. The authors have demonstrated a significantly higher density of stretchable organic transistor arrays with shorter channel lengths, lower operating voltages, and superior on/off ratios compared to the current state-of-the-art intrinsically stretchable organic transistor arrays.

Overall, this work represents a substantial advancement in the realm of stretchable electronic devices. I recommend its publication in Nature Communications, pending certain necessary revisions.

R: We sincerely appreciate the reviewer's carefully reading, valuable comments and constructive suggestions, which are essentially important to improve our manuscript. We carefully revised the manuscript and addressed all the concerns of the reviewer in the revised manuscript. The point-to-point answers for reviewer's comments are shown as follows.

C1: *It remains unclear why the authors chose to use a LiF layer in their approach. It would greatly enhance the manuscript if the authors could provide experimental data elucidating the underlying mechanisms of this choice.*

R1: We sincerely appreciate this reviewer for pointing out the chosen reason of LiF layer. Here, the LiF layer served as a buffer layer that could insulate the contact of CNTs with the photoresist or Si-substrate, facilitating the subsequent lift-off process. Second, the use of LiF sacrificial layer make it easily to transfer the CNTs from the Si-substrate to the elastic substrate through dry method due to the large membrane stress from the evaporation, ultra-high shear modulus (*J. Am. Chem. Soc.* 2012, 134, 37, 15476–15487.) and high surface energy to accommodate deformation during transfer (Supplementary Fig. 5 and Table 1). The similar interface design for electrode has been reported and confirmed to obtain the high interfacial energy in the recent literatures (*Nat. Energy* 2022, 5, 386–397.; *Nature* 2023. DOI: s41586-023-06653-w; *Nat. Commun.* 2023, 14, 4018.). Third, the LiF molecular as a typical electron transport layer would not influence the device performance, thus is available to achieve the aim as the desirable sacrifice layer. To elucidate the mechanisms, we carefully consulted the relevant references and found that the possible mechanism between LiF and CNTs (modified -COOH) may be the Li-O dipole interaction which could mostly reform in a brief time and achieve the reversible breakage and reformation for the deformation during the transfer (*Nat Commun.* 2022. 13, 2279.; *J. Am. Chem. Soc.* 2023, 145, 6, 3526–3534.). As suggested, the detailed mechanism about the interfacial effect is challenging to confirm and deserve to be investigated in future. However, to address the reviewer's concern, we here provided the possible reasons in the revised manuscript (Page 6, line 18 to 22).

C2: *With the exception of the LiF interface technique, the materials used in this study appear to be almost same with those reported in a previous reference*

(<https://www.science.org/doi/10.1126/science.aah4496>). Given the similarity in the fabrication procedures, it is essential for the authors to present additional characterization data of the semiconducting and dielectric layers. Supplementary Table 2 in this manuscript is almost identical to Table S3 in above reference. This is a little bit weird.

R2: We fully agree with the reviewer's comments and apologize for the omission. Accurately, DPP-BT/SEBS-1221 blends and SEBS-H1052 were used as the semiconducting and dielectric layer. The DPP-BT semiconductor have different side chains with the reported DPP-TT semiconductor in the reference (*Science* 2017, 355,59-64.). The DPP-BT semiconductor with the number-averaged molecular weight of 68.4 kDa and polydispersity of 2.4 D was synthesized from our laboratory. We have provided the characterization of DPP-BT semiconductor in the Supplementary Information (Fig. S24). In this revised manuscript, we have added the relevant explanation (Methods, Page18, line 8 and 9).

Figure R12. H NMR spectrum of DPP-BT in 1,1,2,2-tetrachloroethane-d₂ at 100°C.

As for the dielectric layer, we employed the identical SEBS-H1052 as the insulating layer with the same thickness of 1200 nm with the previously reported reference (Jie Xu, et.al. *Science* 2017, 355,59-64.). Indeed, we cast about measuring the capacitance under different strains using various methods, but all failed because it is still challenging to characterize the SEBS dielectric layer with high viscosity, elasticity and small initial thickness of 1200 nm by using the microscale electrodes. In particular, the same materials including the dielectric layer and CNT electrodes have a small deviation in the Poisson's ratio and dielectric constant. Similar experiment results or trend were observed in our previous work with a dielectric layer of 2000 nm and large electrode pairs (*Nat. Commun.* 2022, 13, 7163.). Therefore, we here cited the data in the reported

reference (Jie Xu, et.al. Science 2017, 355,59-64.) to calculate the mobility of our stretchable transistors (Table S3). To avoid any misleading to the reader, we added the relevant explanation in the revised manuscript (Methods, Page18, line 20 and 21).

C3: *To gain a deeper understanding of the low operating voltage and high operational stability observed in the study, it is imperative that the authors provide insights into the underlying mechanisms. Furthermore, it would be valuable to explore whether the use of LiF contributes to improved stretching behaviors of CNT electrodes and, if so, the rationale behind this improvement should be addressed.*

R3: We are grateful for the reviewer's helpful comments and constructive suggestions. In fact, scaling the channel length generally induced the low operating voltage and high operational stability that were also observed in the reported literatures (*Nature* 2018, 555, 83–88; *Science* 2021, 373,88-94). That could be attributed to the enhanced modulation of gate, reduced leakage current and improved interface contact in our short-channel transistors, respectively. Accordance to the reviewer's suggestion, we provided the corresponding explanation in the revised manuscript (Page 8, line 9 to 12).

As for the effect of LiF layer, we further compared the stretchability of transferred micropatterned CNT electrodes with the direct-fabricated CNT electrodes. Unfortunately, there was no obvious improvement on the stretchability and energy level of CNT electrode that may be contributed to the thin LiF layer after transfer onto the elastic substrate. In the revised manuscript, we have added the figure (Fig. S4b) and some sentences to explain this issue (Page 6, line 13 to 15).

Figure R13. Normalized resistance changes as a function of strain for the ordinary CNT electrodes and high-resolution stretchable CNT electrodes after the transfer assisted by LiF.

C4: *The manuscript showed the demonstration of a stretchable active matrix for X-ray imaging. However, it lacks a detailed description of the actual device structure and the fabrication process.*

R4: We thank this reviewer for the insightful comments, and we apologize for the confusion. Actually, we here conducted the main imaging demonstrations by single-pixel imaging system. The device structure and the fabrication process have been provided in Fig 1a, 2a and Supplementary Fig. 1. More details can be seen in the Methods. Unfortunately, limited to the difficult interconnection and reading technology, we did not achieve the stretchable active matrix for dynamic X-ray imaging. However, to address the review's concern, we supplemented the detailed imaging experiment by reading the device one by one and realized a simple imaging demonstration based on

flexible transistor arrays in details. As the figure shown below, the hole can be clearly imaged when the detector array was bent and inserted into the pipe. However, the poor resolution or contrast was observed when placing the detector array outside the pipe, resulted from the larger attenuation as the X-ray irradiation need penetrate two layers to the detectors. To obtain the similar quality of image, rigid flat-panel detectors need even higher irradiation. We have added the figure (Fig. S23) and the relevant explanation in this revised manuscript (Page 10, line 7 to 12 and Page 20, line 13 to 14).

Figure R14. Schematic (left) and experimental results (right) of the hole of the pipe in the wall imaging by the stretchable detectors from inside **(a)** and the rigid detectors from outside **(b)** (All images were taken without noise subtraction).

REVIEWER COMMENTS

Reviewer #1 (Remarks to the Author):

The authors have addressed most of my comments and clarified the doubts. I can recommend publication of this manuscript in Nature Communications

Reviewer #2 (Remarks to the Author):

Thank you for consistently addressing my concerns and revising the manuscript. In fact, I don't have many comments or criticisms regarding the novelty or significance of the work, and I don't intend to create difficulties for the authors. However, I do believe that publishing this manuscript, especially in a prestigious journal like Nature Communications, carries significant risks.

1. My primary concern remains the validity of the excessively high sensitivity, which is orders of magnitude larger than the theoretical value. The explanation of photoconductive gain is not substantiated.

The author repeatedly emphasizes in the response (R1, R2) that, in the context of the X-ray detector based on organic field-effect transistors (OFETs) in this paper, charge transport only occurs within a few nanometers at the interface of the dielectric layer, not within the organic layer. However, this understanding is based on scenarios involving electrical OFET devices or visible light OFET detectors and may not be comprehensive enough for X-ray detection scenarios.

As is well known, X-rays possess strong ionization capabilities, and even insulating materials can be ionized to release free charges when exposed to X-rays. Regardless of how the device structure is designed, all materials near the device electrodes, especially the source and drain electrodes, will be ionized by X-rays. In such cases, applying an electric field between the source and drain electrodes will result in the directed movement of charges under the influence of a strong electric field, generating a current that can be collected. All of this is attributed to the strong ionization capability of X-rays and the ultra-strong electric field between the source and drain electrodes.

Given the author's inaccurate physical image, they assume that the directed movement of charges occurs only in a very small area. Based on this assumption, the author calculates the X-ray sensitivity of the detector, which undoubtedly severely underestimates the actual cross-sectional area during charge transport, leading to a significant overestimation of X-ray sensitivity.

In the response (R1), the author attributes this ultra-high sensitivity to low dark current, a high electric field, a low dose rate, and the photoconductive gain effect, which I disagree with. Firstly, dark current and X-ray sensitivity are two different performance parameters, they actually often conflict with each other. Secondly, the electric field in this paper is only two orders of magnitude larger than in other previous works, making it unreasonable to boost

X-ray sensitivity by a hundred thousand or even a million times. Thirdly, the observed "high X-ray sensitivity at low dose rates" from the results referenced by the author lacks theoretical support. If sensitivity significantly varies with dose rate, it implies poor linearity and limited dynamic range, making it not much useful in actual applications.

In my understanding, this phenomenon arises from the response signal of the air. As the X-ray dose rate increases, the electrode area of the device becomes insufficient to fully collect the charges in the air, leading to saturation.

Finally, the author's proposed photoconductive gain effect remains a typical circular reasoning rather than evidence. The author lacks an understanding of the physical origin of photoconductive gain. Photoconductive gain (G) can be expressed mathematically as the ratio of the photoexcited carrier lifetime to the carrier transit time. The carrier transit time is the time it takes for a carrier to traverse the device under the influence of an applied electric field. If the author wishes to calculate the gain, it should be measured from the above perspective rather than using an unreasonable (overestimated) photocurrent divided by its theoretical limit. This is clearly a form of circular reasoning and cannot provide any explanation for the overestimated photocurrent value.

2. I strongly believe the primary cause of the overestimated sensitivity is the air ionization effect, which is particularly severe in such extremely small devices.

Regarding the air response current results in the response (Fig.R7), I still have reservations about the accuracy of this test. Additionally, the author has not provided detailed information about the test conditions and a schematic diagram of the testing system. After all, even the placement of probes and metal electrodes in the test system can influence the results significantly.

Recently, researchers (<https://doi.org/10.1021/acseenergylett.3c02399>) conducted a thorough investigation into the contribution of air to the sensitivity of X-ray detectors. The study concluded that detectors with coplanar electrodes and a small effective area (Fig. 3d) should be avoided, as this structure is highly susceptible to the influence of air signals, leading to a significant overestimation of X-ray sensitivity. This device structure is almost identical to the one in this paper. Under conditions where the electric field strength is only $0.15 \text{ V}/\mu\text{m}$ and the X-ray dose rate is approximately $30 \mu\text{Gy}/\text{s}$, a response current of about $3 \times 10^{-10} \text{ A}$ was obtained (Fig. 3f and 3e), which closely resembles the response current in Fig.3a of this paper. These findings lead me to maintain reservations about the author's air X-ray response test results (Fig.R7).

I propose two testing approaches to address this issue. The first is to test the detector in a vacuum, which can minimize the impact of air (as similar as the results in Fig. 3f). The second is to use a Pb mask to precisely define your illumination area under X-rays. Otherwise, I cannot be persuaded by the extremely high value of sensitivity presented in this manuscript.

Reviewer #3 (Remarks to the Author):

The authors addressed my comments and made appropriate revisions. Given the high quality of their work, I recommend the publication of this manuscript in Nature Communications.

Responses (R) to the Comments (C)

Responses to the reviewer 1

C: *The authors have addressed most of my comments and clarified the doubts. I can recommend publication of this manuscript in Nature Communications.*

R: We sincerely appreciate the reviewer's positive and insightful comments, which are of crucial significance to our manuscript. Thanks again for your careful reading, valuable comments and helpful suggestions.

Responses to the reviewer 2

Thank you for consistently addressing my concerns and revising the manuscript. In fact, I don't have many comments or criticisms regarding the novelty or significance of the work, and I don't intend to create difficulties for the authors. However, I do believe that publishing this manuscript, especially in a prestigious journal like Nature Communications, carries significant risks.

R: We sincerely appreciate the reviewer's positive, insightful and valuable comments, which are of crucial significance to further improve our manuscript. In fact, the device sensitivity was not our main novelty and had a small influence on the conclusion of this work. Specially, the present X-ray detectors whether in practical applications or in the most reported literatures were measured in the air. However, we fully understand the reviewer's concern about the contribution of the air ionization to the X-ray photocurrent. To address the reviewer's concerns, we have repeated the experiments and remeasured the photoresponse behaviors under X-ray irradiation in a vacuum (<100 Pa). According to the reviewer's suggestions, we also recalculated the photoconductive gain and sensitivity values, which might address all the issues raised by the reviewer. For the reviewer's convenience, the details are included in the point-to-point answers to the comments below.

C1: *My primary concern remains the validity of the excessively high sensitivity, which is orders of magnitude larger than the theoretical value. The explanation of photoconductive gain is not substantiated.*

The author repeatedly emphasizes in the response (R1, R2) that, in the context of the X-ray detector based on organic field-effect transistors (OFETs) in this paper, charge transport only occurs within a few nanometers at the interface of the dielectric layer, not within the organic layer. However, this understanding is based on scenarios involving electrical OFET devices or visible light OFET detectors and may not be comprehensive enough for X-ray detection scenarios.

As is well known, X-rays possess strong ionization capabilities, and even insulating materials can be ionized to release free charges when exposed to X-rays. Regardless of how the device structure is designed, all materials near the device electrodes, especially the source and drain electrodes, will be ionized by X-rays. In such cases, applying an

electric field between the source and drain electrodes will result in the directed movement of charges under the influence of a strong electric field, generating a current that can be collected. All of this is attributed to the strong ionization capability of X-rays and the ultra-strong electric field between the source and drain electrodes.

Given the author's inaccurate physical image, they assume that the directed movement of charges occurs only in a very small area. Based on this assumption, the author calculates the X-ray sensitivity of the detector, which undoubtedly severely underestimates the actual cross-sectional area during charge transport, leading to a significant overestimation of X-ray sensitivity.

In the response (R1), the author attributes this ultra-high sensitivity to low dark current, a high electric field, a low dose rate, and the photoconductive gain effect, which I disagree with. Firstly, dark current and X-ray sensitivity are two different performance parameters, they actually often conflicting with each others. Secondly, the electric field in this paper is only two orders of magnitude larger than in other previous works, making it unreasonable to boost X-ray sensitivity by a hundred thousand or even a million times. Thirdly, the observed "high X-ray sensitivity at low dose rates" from the results referenced by the author lacks theoretical support. If sensitivity significantly varies with dose rate, it implies poor linearity and limited dynamic range, making it not much useful in actual applications.

In my understanding, this phenomenon arises from the response signal of the air. As the X-ray dose rate increases, the electrode area of the device becomes insufficient to fully collect the charges in the air, leading to saturation.

Finally, the author's proposed photoconductive gain effect remains a typical circular reasoning rather than evidence. The author lacks an understanding of the physical origin of photoconductive gain. Photoconductive gain (G) can be expressed mathematically as the ratio of the photoexcited carrier lifetime to the carrier transit time. The carrier transit time is the time it takes for a carrier to traverse the device under the influence of an applied electric field. If the author wishes to calculate the gain, it should be measured from the above perspective rather than using an unreasonable (overestimated) photocurrent divided by its theoretical limit. This is clearly a form of circular reasoning and cannot provide any explanation for the overestimated photocurrent value.

R1: We thank this reviewer's carefully reading, insightful comments and valuable suggestions. We apologize for the confusion about the actual cross-sectional area during charge transport. In fact, we fully agree with the reviewer "applying an electric field between the source and drain electrodes will result in the directed movement of charges under the influence of a strong electric field, generating a current that can be collected". Precisely because of this, we assumed that the directed movement region of charges between source and drain electrodes as the effective area, which also was our transistor channel (channel length/width of 2 μm /8 μm) that generally occurred the separation, transport and even combination of photogenerated carriers (Fig. R1). In the reported literatures of resistance- or transistor-based X-ray detectors (*Nat. Commun.* 2016,

7:13063.; *Nat. Commun.* 2020, 11, 2136.; *J. Phys. Chem. Lett.* 2020, 11, 432–437.; *Adv. Sci.* 2020, 7, 2001522.), the actual cross-sectional area was all defined as the effective overlap area between source and drain electrodes.

Figure R1. Lateral and top views of the stretchable short-channel organic transistors.

To address the reviewer’s concern about the sensitivity, we have remeasured the I-V curves under X-ray irradiation in vacuum and air conditions. Actually, we found the air ionization had a small influence on the photocurrent at a low dose rate and an unobvious variation at the high dose rate region (Fig. R2). Noted that the degraded performance (including the on- and off-current values) at dark was caused by the associated changes in series resistance, leakage current and ambient noises from the connected wires and vacuum equipment. In particular, the dark current ($\sim 10^{-12}$ A) increased by 1~2 orders of magnitude than the normal measurement ($10^{-13}\sim 10^{-14}$ A), severely limited the detection limit to 350 nGy s^{-1} in vacuum ($24.34 \text{ nGy}_{\text{air}} \text{ s}^{-1}$ in air). In this manuscript, we have added the relevant explanations (Page 11, Line 12 to 23) and figures (Supplementary Fig. 14).

Figure R2. Photoresponse comparison of stretchable short-channel organic transistors from different batches under X-ray irradiation in vacuum (<100 Pa) and air condition.

From the results, we confirmed the response deviation at low dose rates and further remeasured the photoresponse behaviors under different dose rates in a vacuum (<100 Pa), as shown below (Fig. R3). According to the reviewer’s suggestions, we recalculated the sensitivity value of $5.74 \times 10^6 \mu\text{C Gy}^{-1} \text{ cm}^{-2}$, which was corresponding to the high gain value ($G = \tau/t = 4.27 \times 10^6$) at a strong electric field provided by our short-channel transistors. Additionally, the photocurrent showed well linear behaviors at a vacuum that was consistent with the reviewer’s comment. Indeed, the present X-ray detectors whether in practical applications or in the most reported literatures were measured in the air. Specially, the transistor performance was largely influenced by the connected wires and vacuum equipment, thereby might not reveal the actual detection limit. Therefore, we also retained the photoresponse results measured in the air with the

marked subscript for discrimination. However, to address the reviewer's concern, we have modified the relevant figures (Fig. 3d, e and Supplementary Fig. 15) and explanations (Page 11, Line 17 to 23) of the sensitivity in accordance with the results under the vacuum condition in the revised manuscript.

Figure R3. Photoresponse curves and current density of stretchable short-channel organic transistors under different dose rates in a vacuum (<100 Pa).

As for the photoconductive gain (G), we have noticed the two calculating approaches, the first is the ratio of the photoexcited carrier lifetime to the carrier transit time (*Nat. Commun.* 2016, 7:13063.; *Nat. Commun.* 2020, 11, 2136.) and the other is calculated by the current ratio of the photoinduced and theoretical photocurrent (*Nat. Commun.* 2016, 7:13063.; *Nat. Photon.* 2017, 11, 726–732.;). The latter was also referred to as the charge collection efficiency (CCE) (*J. Phys. Chem. Lett.* 2020, 11, 432–437.). Typically, it is difficult to accurately measure the carrier lifetime and transit time of semiconductors, thereby the current ratio calculating approach was adopted in most literatures. However, to address the reviewer's concern, we remeasured and fitted the characteristic decay time after the X-ray irradiation to obtain the photoexcited carrier lifetime (Fig. R4) (*Adv. Mater.* 2018, 1802883.; *Nat. Commun.* 2020, 11, 2136.; *Nat. Commun.* 2021, 12, 1798.). Referring to the reported literatures (*Nat. Commun.* 2016, 7:13063.; *Nat. Commun.* 2020, 11, 2136.), we further calculated the transit time of organic semiconductor for hole transport as determined from field effect transistor measurements. As a result, we obtained a high photoconductive gain about 4.27×10^6 ($\tau = 0.0769$ s, $t = 18.0$ ns). We have revised the relevant calculation and explanations of photoconductive gain in the revised manuscript (Page 11, Line 14) and Supplementary information (Part 3, Page 13 and Supplementary Fig. 16).

The expression of gain (G) is depicted as follows:

$$G = \frac{\tau}{t}$$

where τ and t are the photoexcited carrier lifetime and carrier transit time, respectively.

The transit time of the hole carriers along the channel:

$$t = \frac{L^2}{\mu V}$$

where L is the channel length. The μ and V are the carrier mobility and source/drain voltage, respectively.

The decay trends of the photonic transistors under different dose rates were calculated

using fitted by a double exponential function and average time constant, τ values, were calculated by the equations:

$$I = A_1 e^{-t/\tau_1} + A_2 e^{-t/\tau_2} + B$$

$$\tau_{\text{average}} = \frac{\sum_i^n A_i \tau_i}{\sum_i^n A_i}$$

Figure R4. Experimental and fitted curves of the response of the short-channel intrinsically stretchable organic transistor for different dose rates of X-ray radiation.

C2: I strongly believe the primary cause of the overestimated sensitivity is the air ionization effect, which is particularly severe in such extremely small devices.

Regarding the air response current results in the response (Fig.R7), I still have reservations about the accuracy of this test. Additionally, the author has not provided detailed information about the test conditions and a schematic diagram of the testing system. After all, even the placement of probes and metal electrodes in the test system can influence the results significantly.

Recently, researchers (<https://doi.org/10.1021/acseenergylett.3c02399>) conducted a thorough investigation into the contribution of air to the sensitivity of X-ray detectors. The study concluded that detectors with coplanar electrodes and a small effective area (Fig. 3d) should be avoided, as this structure is highly susceptible to the influence of air signals, leading to a significant overestimation of X-ray sensitivity. This device structure is almost identical to the one in this paper. Under conditions where the electric field strength is only $0.15 \text{ V}/\mu\text{m}$ and the X-ray dose rate is approximately $30 \mu\text{Gy/s}$, a response current of about $3 \times 10^{-10} \text{ A}$ was obtained (Fig. 3f and 3e), which closely resembles the response current in Fig.3a of this paper. These findings lead me to

maintain reservations about the author's air X-ray response test results (Fig.R7). I propose two testing approaches to address this issue. The first is to test the detector in a vacuum, which can minimize the impact of air (as similar as the results in Fig. 3f). The second is to use a Pb mask to precisely define your illumination area under X-rays. Otherwise, I cannot be persuaded by the extremely high value of sensitivity presented in this manuscript.

R2: Thank you very much for your comments and suggestions. We fully agree with the reviewer's comment and understand all concerns. In fact, the air X-ray response results (previous Fig.R7) were tested and calculated according to the reviewer's suggestions with an average ionized air cross-sectional area of 1 mm^2 at a high dose rate. However, to address the reviewer's concern, we remeasured the X-ray response in a vacuum ($<100 \text{ Pa}$) and air with a sealed chamber (Fig. R5). To avoid any possible influences from the probes and metal electrodes, the peripheral areas, probes and wires were fully shielded in the X-ray measurement. In this manuscript, we have added the relevant explanations (Page 11, Line 15 to 17) and figures (Supplementary Fig. 17).

Figure R5. Schematic diagrams of the vacuum measurement equipment and the connected stretchable devices (The device and wires were not fully shielded for intuitive description).

As mentioned about the X-ray measurements in a vacuum, we agree the reviewer that it's an effective approach to avoid the influence of air and provide more valid sensitivity. According to the reviewer's suggestion, we remeasured and recalculated the sensitivity using the vacuum condition rather than the air environment (Fig. R6). The calculated sensitivity was $5.74 \times 10^6 \mu\text{C Gy}^{-1} \text{ cm}^{-2}$, that was relatively reasonable and reliable at a high electric field of $2.5 \text{ V}/\mu\text{m}$. Unfortunately, our device must be connected with the vacuum measurement equipment by three wire leads, which induced the degraded performance. Additionally, the noise caused by the vacuum pump greatly affected the dark current and detection limit of our short-channel device. In particular, the present X-ray detectors whether in practical applications or in the most reported literatures were measured in the air. Therefore, considering other reviewer's comments, we also retained the photoresponse results measured in the air. In this revised manuscript, we have revised the relevant figures (Fig. 3d and Supplementary Fig. 15) and explanations (Page 11, Line 17 to 23) about the sensitivity to better clarify the actual device performance.

Figure R6. Photoresponse curves and current density of stretchable short-channel organic transistors under different dose rates in a vacuum (<100 Pa).

Responses to the reviewer 3

C: *The authors addressed my comments and made appropriate revisions. Given the high quality of their work, I recommend the publication of this manuscript in Nature Communications.*

R: We sincerely appreciate the reviewer's positive and valuable comments, which are of crucial significance to further improve our manuscript. Thanks again for your careful reading, insightful comments and helpful suggestions.

REVIEWERS' COMMENTS

Reviewer #2 (Remarks to the Author):

1. I appreciate the author's efforts in measuring the sensitivity of X-ray detection, including experiments conducted in a vacuum environment. However, further discussion is needed regarding the test results of this experiment. In the response, the author retested the X-ray responsiveness of the detector under vacuum and air conditions separately. At an X-ray dose rate of 1.29 mGy/s, the detector's response currents in vacuum and air were similar, with the signal in vacuum even surpassing that in air. However, at a dose rate of 30.6 uGy/s, the signal in vacuum was approximately 30% of the signal in air. It can be observed that the dominant air signal at low dose rates mysteriously disappears at high dose rates, and the author does not provide an explanation for this phenomenon. Based on the experimental results, I speculate that unexpected ionization may still occur between the source and drain electrodes of the detector under conditions of 1.29 mGy/s X-ray dose rate in vacuum. This phenomenon generates sufficient free charges to cause saturation effects. In this scenario, even with the presence of air, free charges originated from the air cannot be effectively collected by the electrodes, resulting in similar response currents in both vacuum and air conditions.

Regarding the origin of these free charges, I have two hypotheses. First, even in the absence of air, the possibility of ionization of CNT electrodes should be considered. As known, CNTs have a significantly lower work function than metals and are often used as field-emission electrode materials in cathode-ray tubes, exhibiting excellent electron emission performance. Therefore, even in a vacuum, surface electrons of CNT electrodes can be ionized by X-rays, forming free charges that generate response currents in high electric fields. Second, it is possible that due to insufficient vacuum levels (<100Pa) in the author's equipment, residual air could still be ionized into free charges at higher X-ray dose rates. Additionally, due to the small electrode width and narrow channel of the detector, even a small number of ionized free charges are sufficient to induce saturation effects in the electrodes. Indeed, the crucial aspect of a sensitivity measurement is the accurate definition of the actual cross-sectional area. However, the author insists on using $2\mu\text{m} \times 8\mu\text{m}$ as the current cross-sectional area to evaluate the sensitivity in the manuscript. Nevertheless, based on the previous discussion, this area ($16\mu\text{m}^2$) may be much smaller than the actual cross-sectional area when free charge motion occurs under high electric fields.

Indeed, the most reliable measurement involves using a thick metal mask (such as Pb) to cover all areas except the conduction channel. However, I acknowledge the considerable challenge in creating such a mask due to the extremely small size of your device. Considering that the primary innovation of this paper does not hinge on achieving exceptionally high performance, I would like to propose the following suggestions:

(1) When articulating the X-ray response capability of the detector in the manuscript, avoid employing traditional X-ray sensitivity as the metric. Instead, consider presenting your device using X-ray photocurrent rather than current density. For example, $(X\text{-ray current}(A))/(Dose$

rate(Gy/s)).

(2) Discuss potential sources contributing to the detector's response signal in the manuscript, such as organic photoactive layers, ionized air, and CNT electrode emission. These factors make the accurate determination of the current cross-sectional area nearly impossible.

2. In the assessment of optical gain, the author obtained a carrier lifetime (τ) value in the range of milliseconds, a clear deviation from the fundamental optical physics principles governing organic semiconductors, which typically operate in the picosecond to nanosecond range. The experiment's details were not thoroughly elucidated, but based on the experimental data, it seems that X-ray excitation was used to test the transient photocurrent (TPC). Theoretically, the result obtained from this experiment is the transit time (t), not the carrier lifetime, which usually necessitates testing with a pulsed light source featuring a very narrow pulse width (on the order of nanoseconds or less, much shorter than the photocurrent transient time. I assume that the X-ray source utilized in this study cannot produce such a narrow pulse. Consequently, the prolonged decay in your measured current is likely due to the inability of your light source to turn off rapidly. It actually measures the decay of the light source not your device. The most common practice to measure τ is through measuring the transient photoluminescence lifetime (TRPL lifetime) of your organic material, which can approximate the carrier lifetime. Considering the mobility of the materials (which can approximate the transit time t , as the author did), and a typical PL lifetime of organic semiconductors, one can simply estimate the photoconductive gain, which should not reach the levels reported in this paper.

Given my earlier advice to the author against directly reporting X-ray sensitivity, there is no need to provide a detailed explanation for the calculated high X-ray sensitivity. Therefore, I recommend that the author exclude the experimental and computational sections related to optical gain from the manuscript and avoid underscoring the contribution of optical gain to the X-ray response of the detector. The measurement and discussion of photoconductive gain are not accurate

Overall, I can acknowledge that the primary technical contribution of the paper might not be an exceptionally high-performance X-ray detector. However, since you are presenting a paper on an X-ray detector, sensitivity remains a crucial parameter, and its accuracy is of utmost importance to your readers when it is published.

One more comment regarding your title, I presume the primary contribution of the paper lies in a fabrication method for creating a TFT array potentially useful for X-ray imaging. Yet, when reporting the X-ray image resolution, you actually utilize a single-pixel device. In this context, I would like to note that the term "transistor arrays and high-resolution X-ray imaging" could be somewhat misleading. The image resolution reported in the paper is unrelated to the transistor arrays developed in this study.

Responses (R) to the Comments (C)

Responses to the reviewer 2

C1: *I appreciate the author's efforts in measuring the sensitivity of X-ray detection, including experiments conducted in a vacuum environment. However, further discussion is needed regarding the test results of this experiment. In the response, the author retested the X-ray responsiveness of the detector under vacuum and air conditions separately. At an X-ray dose rate of 1.29 mGy/s, the detector's response currents in vacuum and air were similar, with the signal in vacuum even surpassing that in air. However, at a dose rate of 30.6 uGy/s, the signal in vacuum was approximately 30% of the signal in air. It can be observed that the dominant air signal at low dose rates mysteriously disappears at high dose rates, and the author does not provide an explanation for this phenomenon. Based on the experimental results, I speculate that unexpected ionization may still occur between the source and drain electrodes of the detector under conditions of 1.29 mGy/s X-ray dose rate in vacuum. This phenomenon generates sufficient free charges to cause saturation effects. In this scenario, even with the presence of air, free charges originated from the air cannot be effectively collected by the electrodes, resulting in similar response currents in both vacuum and air conditions.*

Regarding the origin of these free charges, I have two hypotheses. First, even in the absence of air, the possibility of ionization of CNT electrodes should be considered. As known, CNTs have a significantly lower work function than metals and are often used as field-emission electrode materials in cathode-ray tubes, exhibiting excellent electron emission performance. Therefore, even in a vacuum, surface electrons of CNT electrodes can be ionized by X-rays, forming free charges that generate response currents in high electric fields. Second, it is possible that due to insufficient vacuum levels (<100Pa) in the author's equipment, residual air could still be ionized into free charges at higher X-ray dose rates. Additionally, due to the small electrode width and narrow channel of the detector, even a small number of ionized free charges are sufficient to induce saturation effects in the electrodes. Indeed, the crucial aspect of a sensitivity measurement is the accurate definition of the actual cross-sectional area. However, the author insists on using $2\mu\text{m} \times 8\mu\text{m}$ as the current cross-sectional area to evaluate the sensitivity in the manuscript. Nevertheless, based on the previous discussion, this area ($16\mu\text{m}^2$) may be much smaller than the actual cross-sectional area when free charge motion occurs under high electric fields.

Indeed, the most reliable measurement involves using a thick metal mask (such as Pb) to cover all areas except the conduction channel. However, I acknowledge the considerable challenge in creating such a mask due to the extremely small size of your device. Considering that the primary innovation of this paper does not hinge on achieving exceptionally high performance, I would like to propose the following suggestions:

(1) When articulating the X-ray response capability of the detector in the manuscript, avoid employing traditional X-ray sensitivity as the metric. Instead, consider presenting

your device using X-ray photocurrent rather than current density. For example, (X-ray current(A))/(Dose rate(Gy/s)).

(2) Discuss potential sources contributing to the detector's response signal in the manuscript, such as organic photoactive layers, ionized air, and CNT electrode emission. These factors make the accurate determination of the current cross-sectional area nearly impossible.

R1: We sincerely appreciate the reviewer's positive, insightful and valuable comments, which are of crucial significance to further improve our manuscript. In fact, we fully agree with the reviewer's hypotheses about the origin of these free charges. According to the reviewer's suggestions, we have added the relevant explanations about the possible contribution from the ionization of CNT electrodes and less air to the response signal (Page 9, Line 6 to 8).

As for the current density, we have modified the relevant figures (Fig. 3d, f, h and i and Supplementary Fig. 17) and explanations (Page 9, Line 4, 9, 11, 18 and 24) in this manuscript. However, considering the sensitivity calculation of most reported literatures based on current density, we reserved the sensitivity metric and modified the emphasis on photosensitivity performance (Page 12, Line 15 and 16).

Figure R1. Ultrasensitive detection performance of intrinsically stretchable transistor-based organic X-ray image sensors.

Figure R2. Device-to-device variation of X-ray photocurrent in our short-channel stretchable transistor-based detectors.

C2: In the assessment of optical gain, the author obtained a carrier lifetime (τ) value in the range of milliseconds, a clear deviation from the fundamental optical physics principles governing organic semiconductors, which typically operate in the picosecond to nanosecond range. The experiment's details were not thoroughly elucidated, but based on the experimental data, it seems that X-ray excitation was used to test the transient photocurrent (TPC). Theoretically, the result obtained from this experiment is the transit time (t), not the carrier lifetime, which usually necessitates testing with a pulsed light source featuring a very narrow pulse width (on the order of nanoseconds or less, much shorter than the photocurrent transient time). I assume that the X-ray source utilized in this study cannot produce such a narrow pulse. Consequently, the prolonged decay in your measured current is likely due to the inability of your light source to turn off rapidly. It actually measures the decay of the light source not your device. The most common practice to measure τ is through measuring the transient photoluminescence lifetime (TRPL lifetime) of your organic material, which can approximate the carrier lifetime. Considering the mobility of the materials (which can approximate the transit time t , as the author did), and a typical PL lifetime of organic semiconductors, one can simply estimate the photoconductive gain, which should not reach the levels reported in this paper.

Given my earlier advice to the author against directly reporting X-ray sensitivity, there is no need to provide a detailed explanation for the calculated high X-ray sensitivity. Therefore, I recommend that the author exclude the experimental and computational sections related to optical gain from the manuscript and avoid underscoring the contribution of optical gain to the X-ray response of the detector. The measurement and discussion of photoconductive gain are not accurate

Overall, I can acknowledge that the primary technical contribution of the paper might not be an exceptionally high-performance X-ray detector. However, since you are presenting a paper on an X-ray detector, sensitivity remains a crucial parameter, and its accuracy is of utmost importance to your readers when it is published.

One more comment regarding your title, I presume the primary contribution of the paper lies in a fabrication method for creating a TFT array potentially useful for X-ray imaging. Yet, when reporting the X-ray image resolution, you actually utilize a single-

pixel device. In this context, I would like to note that the term "transistor arrays and high-resolution X-ray imaging" could be somewhat misleading. The image resolution reported in the paper is unrelated to the transistor arrays developed in this study.

R2: Thank you very much for your comments and suggestions. We fully agree with the reviewer's comment and understand all concerns. As the reviewer suggested, we have deleted the relevant calculation, figure and explanations of photoconductive gain in the revised manuscript (Page 8, Line 9 and Page 9, Line 11 and 12) and Supplementary information (Part 3, Page 13 and Supplementary Fig. 16).

As for the title, given the imaging experiments based on both the single pixel and transistor matrix, we believe it has no obvious error. In fact, our TFT array demonstrated the excellent imaging capability for next-generation extendable digital imaging (Supplementary Fig. 23). Therefore, considering other reviewers' comments, we reserved the "transistor arrays and high-resolution X-ray imaging".